# Evaluating the assimilation of S5P/Tropomi NRT SO₂ columns and layer height data into the CAMS integrated forecasting system (CY47R1), based on a case study of the 2019 Raikoke eruption

Antje Inness[1], Melanie Ades[1], Dimitris Balis[3], Dmitry Efremenko[2], Johannes Flemming[1], Pascal Hedelt[2], Maria-Elissavet Koukouli[3], Diego Loyola[2], and Roberto Ribas[1]

[1]European Centre for Medium-Range Weather Forecasts (ECMWF), Shinfield Park, Reading, RG2 9AX, UK.
[2]Deutsches Zentrum für Luft und Raumfahrt (DLR), Institut für Methodik der Fernerkundung (IMF), Oberpfaffenhofen, Germany.
[3]Laboratory of Atmospheric Physics, Aristotle University of Thessaloniki, Greece

*Correspondence to*: Antje Inness (antje.inness@ecmwf.int)

**Abstract.**

The Copernicus Atmosphere Monitoring Service (CAMS), operated by the European Centre for Medium-Range Weather Forecasts on behalf of the European Commission, provides daily analyses and 5-day forecasts of atmospheric composition, including forecasts of volcanic sulphur dioxide (SO₂) in near-real time. CAMS currently assimilates total column SO₂ products from the GOME-2 instruments on MetOp-B and -C and the TROPOMI instrument on Sentinel-5P which give information about the location and strength of volcanic plumes. However, the operational TROPOMI and GOME-2 data do not provide any information about the height of the volcanic plumes and therefore some prior assumptions need to be made in the CAMS data assimilation system about where to place the resulting SO₂ increments in the vertical. In the current operational CAMS configuration, the SO₂ increments are placed in the mid-troposphere, around 550 hPa or 5 km. While this gives good results for the majority of volcanic emissions, it will clearly be wrong for eruptions that inject SO₂ at very different altitudes, in particular exceptional events where part of the SO₂ plume reaches the stratosphere.

A new algorithm, developed by DLR for GOME-2 and TROPOMI and optimized in the frame of the ESA-funded Sentinel-5P Innovation–SO₂ Layer Height Project, the Full-Physics Inverse Learning Machine (FP_ILM) algorithm, retrieves SO₂ layer height from TROPOMI in NRT in addition to the SO₂ column. CAMS is testing the assimilation of these products, making use of the NRT layer height information to place the SO₂ increments at a retrieved altitude. Assimilation tests with the TROPOMI SO₂ layer height data for the Raikoke eruption in June 2019 show that the resulting CAMS SO₂ plume heights agree better with IASI plume height data than operational CAMS runs without the TROPOMI SO₂ layer height information and that making use of the additional layer height information leads to improved SO₂ forecasts than when using the operational CAMS configuration. Including the layer height information leads to higher modelled TCSO₂ values in better agreement with the satellite observations. However, the plume area and SO₂ burden are generally overestimated in the CAMS analysis also when LH data are used. The main reason for this overestimation is the coarse horizontal resolution used in the minimisations. By assimilating the SO₂ layer height data the CAMS system can predict the overall location of the Raikoke SO₂ plume up to 5 days in advance for about 20 days after the initial eruption, which is better than with the operational CAMS configuration (without prior knowledge of the plume height) where the forecast skill reduces much more for longer forecast lead-times.

## 1 Introduction

Volcanoes can cause serious disruptions for society, not just for people living near them, but also further afield when ash and sulphur dioxide ($SO_2$) emitting from highly explosive eruptions reach the upper troposphere or stratosphere, above the clouds, and therefore are transported over vast distances by the prevailing winds. Ash and $SO_2$ are a serious concern for the aviation industry, reducing visibility and in severe cases can lead to engine failure or cause permanent damage to aircraft engines (Prata et al., 2019). The immediate danger to the aircraft comes mainly from the emitted ash, although $SO_2$ is also an

aviation hazard, potentially causing long-term damage via corrosion and sulfidation of the engines (Schmidt et al., 2014). If sufficient SO2 is diffused into the aircraft cabin this could potentially lead to respiratory problems for passengers and crew. Planes therefore try to avoid volcanic plumes and after the 2010 eruption of the Icelandic Eyjafjallajökull volcano (e.g. Stohl et al., 2011; Dacre et al., 2011; Thomas and Prata, 2011) European air traffic was grounded for several days. Forecasts of the location and the altitude of volcanic $SO_2$ or ash plumes can therefore provide important information for the aviation industry.

Satellite retrievals of volcanic ash and $SO_2$ can help to track volcanic plumes, as done by the Support to Aviation Control Service (sacs.aeronomie.be; Brenot et al., 2014) and the EUNADICS (European Natural Airborne Disaster Information and Coordination System for Aviation) prototype Early Warning System (Brenot et al., 2021). These services, as well as plume dispersion modelling (e.g. de Leeuw et al., 2021; Harvey and Dacre, 2016), are used by the Volcanic Ash Advisory Centres (VAACs) to advise civil aviation authorities in case of volcanic eruptions. While $SO_2$ is often used as a proxy for ash, the

$SO_2$ and ash plumes can be located at different altitudes and be transported in different directions as was the case for the Icelandic Grímsvötn eruption in 2011 (Moxnes et al., 2013, Prata et al., 2017).

The Copernicus Atmosphere Monitoring Service (CAMS), operated by the European Centre for Medium-Range Weather Forecasts (ECMWF) on behalf of the European Commission, provides daily analyses and 5-day forecasts of atmospheric

composition, including forecasts of volcanic $SO_2$ in near-real time (NRT). Since the CAMS forecast system runs within 3 hours of the observations being taken, information about volcanic $SO_2$ emission strength and the altitude of $SO_2$ plumes is usually not available, with only the total column-integrated $SO_2$ amount ($TCSO_2$) able to be provided to adjust the model's predictions. CAMS uses the method described in Flemming and Inness (2013) in its operational NRT system to routinely assimilate NRT $TCSO_2$ data from the Global Ozone Monitoring Experiment-2 (GOME-2) instruments produced by

Eumetsat's Satellite Application Facility on Atmospheric Composition Monitoring (ACSAF) and from the Sentinel-5P Tropospheric Monitoring Instrument (TROPOMI) provided by the European Space Agency (ESA). Both products are derived using retrievals developed by the German Aerospace Centre (DLR) and give information about the emitted volcanic $SO_2$ and the horizontal location in NRT but do not provide any information about the altitudes of the volcanic plumes. Prior assumptions therefore need to be made in the CAMS data assimilation system about where in the vertical the resulting $SO_2$

increments should be placed. In the absence of NRT height information, the default is to place the $SO_2$ increments in the mid-troposphere, around 550 hPa or 5 km. Although clearly a simplified approach, the method is a reasonable approximation to the real situation, using the data assimilation procedure as a mid-tropospheric $SO_2$ source in areas of elevated volcanic $TCSO_2$. The $SO_2$ analysis field will then be transported by the model's prevailing winds and thereby result in quite realistic volcanic $SO_2$ plumes. While this method produces good results for a large number of volcanic eruptions that inject $SO_2$ into

the mid-troposphere, it will clearly be wrong for eruptions that inject $SO_2$ at very different altitudes, in particular for the most explosive events where part of the $SO_2$ reaches the stratosphere. In those cases, the CAMS system will not be able to forecast the $SO_2$ transport well, because the model $SO_2$ plume will be located at the wrong altitude where the prevailing winds might transport the $SO_2$ in the wrong direction or height. The availability and use of NRT information about the altitude of the volcanic plumes would greatly improve the quality of the CAMS $SO_2$ analysis and subsequent forecasts.


For hindcasts of volcanic eruptions with a system that does not run in NRT it is easier to make use of better injection height information. In this case, observations about injection height and emission strength might be available. Furthermore, CAMS

can run an ensemble of $SO_2$ tracers emitted at different altitudes and determine the best altitude and emission strength from comparisons of the resulting model fields with the available $TCSO_2$ observations, using a method described in Flemming and Inness (2013). The parameters (plume height and emission flux) derived in this way can subsequently be used to provide a volcanic $SO_2$ source term in the CAMS forecast model and can also be used in the data assimilation system to modify the $SO_2$ background error standard deviation to peak at the corresponding model level. However, this is not possible in NRT.

A new algorithm, developed by DLR for GOME-2 and adapted to TROPOMI, which is currently being optimized in the frame of the ESA-funded Sentinel-5P (S5P) Innovation–$SO_2$ Layer Height Project (S5P+I: $SO_2$LH), the Full-Physics Inverse Learning Machine (FP_ILM) algorithm (Hedelt et al., 2019), retrieves $SO_2$ layer height (LH) information from TROPOMI in NRT in addition to the $SO_2$ column. This is different from the operational ESA NRT TROPOMI product which does not provide plume height information. CAMS is testing the assimilation of the FP_ILM data, making use of the NRT LH information. In this paper we document the current use of the operational $TCSO_2$ data in the CAMS data assimilation system, present results from assimilation tests with the FP_ILM TROPOMI $SO_2$ LH data for the eruption of the Raikoke volcano in June 2019 and show that making use of the NRT LH information leads to improved $SO_2$ analyses and in particular $SO_2$ forecasts.

This paper is structured in the following way. Section 2 describes the $SO_2$ datasets used in this study and Section 3 describes the CAMS model and $SO_2$ data assimilation setup. Section 4 presents the results from the assimilation of TROPOMI data for eruption of Raikoke in June 2019, including sensitivity studies to evaluate choices made for the $SO_2$ background errors, and evaluates the quality of the resulting $SO_2$ analyses and forecasts with and without LH information. Section 5 presents the conclusions.

## 2 Datasets

The $SO_2$ satellite data currently used in the CAMS NRT system are the operational $TCSO_2$ products from TROPOMI on S5P produced by ESA and from the GOME-2 instruments on MetOp-B and MetOp-C produced by Eumetsat's ACSAF. These data come with a volcanic flag, i.e. the data producers mark the pixels that are affected by volcanic $SO_2$, and only pixels that are flagged as volcanic are assimilated in the CAMS system. Using TROPOMI in addition to GOME-2 has two advantages: (1) TROPOMI has better spatial coverage and a lower detection limit than GOME-2 and (2) because TROPOMI has a different overpass time (9.30 UTC for MetOp, 13.30 UTC for S5P) using both instruments improves the chances of having an overpass over a volcano when an eruption happens or shortly afterwards.

### 2.1 NRT TROPOMI $TCSO_2$ data

TROPOMI on board the S5P satellite provides high-resolution spectral measurements in the ultraviolet (UV), visible (Vis), near infrared and shortwave-infrared parts of the spectrum, allowing several atmospheric trace gases to be retrieved, including $SO_2$ from the UV–Vis part of the spectrum. The horizontal resolution of TROPOMI for the UV-Vis is 5.5 km x 3.5 km (7 km x 3.5 km before 6 August 2019) with daily global coverage. The theoretical baseline for the operational TROPOMI $SO_2$ retrieval is described in Theys et al. (2017) and further information can be found in Algorithm Theoretical Basis Document (ATBD), Product User Manual (PUM) and readme files available from the TROPOMI website (http://www.tropomi.eu/documents/). The atmospheric $SO_2$ vertical column density is retrieved in three fitting windows (312–326 nm, 325–335 nm and 360–390 nm) using a Differential Optical Absorption Spectroscopy (DOAS) method (Platt and Stutz, 2008; Platt, 2017), in which the slant $SO_2$ column is retrieved and converted into vertical columns by using air mass factors. The log-ratio of the observed UV-visible spectrum of radiation backscattered from the atmosphere and an

observed reference spectrum are used to derive a slant column density, which represents the $SO_2$ concentration integrated along the mean light path through the atmosphere. This is performed by fitting $SO_2$ absorption cross-sections to the measured reflectance in a given spectral interval. In a second step, slant columns are corrected for possible biases. Finally, the slant columns are converted into vertical columns by means of air mass factors obtained from radiative transfer calculations, accounting for the viewing geometry, clouds, surface properties and prior $SO_2$ vertical profile shapes. A volcano activity detection algorithm going back to Brenot et al. (2014) is used to identify elevated $SO_2$ values from volcanic eruptions (see Table 1). CAMS only assimilates $SO_2$ pixels that have flag values of 1 (enhanced $SO_2$ detection) or 2 (enhanced $SO_2$ detection in vicinity of known volcano). Furthermore, only TROPOMI $SO_2$ pixels with values greater than 5 DU are assimilated in the operational CAMS system to avoid assimilating $SO_2$ from outgassing volcanoes which are covered by $SO_2$ emissions in the CAMS model. The TROPOMI $SO_2$ data are super-obbed, i.e. in a pre-processing step area means are created by averaging all data (observation values as well as errors) in a model grid box to the model resolution (TL511, about 40km). These super-observations are then used in the CAMS system without further thinning.

The DOAS vertical column $SO_2$ retrieval requires an assumption for a prior $SO_2$ profile to convert the slant columns into vertical columns. Since this profile shape is generally not known at the time of the observation and it is also not know whether the observed $SO_2$ is of volcanic origin or from pollution (or both) the TROPOMI algorithm calculates four vertical columns for different hypothetical $SO_2$ profiles. One vertical column is provided for anthropogenic $SO_2$ with the prior $SO_2$ profile taken from the TM5 CTM and three for volcanic scenarios assuming the $SO_2$ is either located in the boundary layer, in the mid-troposphere (around 7 km) or in the stratosphere (around 15 km). These volcanic prior profiles are box profiles of 1 km thickness, located at the corresponding altitudes. The NRT CAMS system uses the mid-troposphere product. TROPOMI $SO_2$ data are provided with averaging kernels based on the prior hypothetical $SO_2$ profiles (i.e. the 1 km box profiles centred around the assumed $SO_2$ altitude for the volcanic columns). However, as these do not provide information about the real altitude of a specific volcanic plume they are not used in the CAMS system. More information about the NRT TROPOMI $SO_2$ retrieval can be found in the TROPOMI ATBD. For the TROPOMI data (and also the other $SO_2$ products used in this paper) observation errors as given by the data providers are used within the CAMS data assimilation system.

| Flag value | Description |
|---|---|
| 0 | No detection |
| 1 | Enhanced $SO_2$ detection |
| 2 | Enhanced $SO_2$ detection in vicinity of known volcano |
| 3 | Enhanced $SO_2$ in vicinity of anthropogenic source |
| 4 | Enhanced $SO_2$ in SAA or for SZA>70° |

**Table 1: Volcanic $SO_2$ flags provided for the TROPOMI $SO_2$ products. The same flags are also used for TROPOMI $SO_2LH$ data and GOME-2C GPD4.9 $SO_2$ data.**

## 2.2 FP_ILM NRT TROPOMI Layer Height data

Hedelt et al. (2019) have developed an algorithm called 'Full-Physics Inverse Learning Machine' (FP_ILM) for the retrieval of the $SO_2$ LH based on Sentinel-5 precursor/TROPOMI data using a coupled Principal Component Analysis and Neural Network approach including regression. This algorithm is an improvement of the original FP_ILM algorithm developed by Efremenko et al. (2017) for the retrieval of the $SO_2$ LH based on GOME-2 data using a Principal Component Regression technique. Recently, this algorithm has also been adapted to retrieve $SO_2$ LH data from the Ozone Monitoring Instrument

(OMI) on the Aura satellite (Fedkin et al., 2021). Furthermore, the FP_ILM algorithm has been used for the retrieval of
ozone profile shapes (Xu et al., 2017) and the retrieval of surface properties accounting for bidirectional reflectance
distribution function effects (Loyola et al., 2020). In general, the FP_ILM algorithm creates a mapping between the spectral
radiance and atmospheric parameter using machine learning methods. The time-consuming training phase of the algorithm
using radiative transfer model calculations is performed off-line, and only the inversion operator has to be applied to satellite
measurements which makes the algorithm extremely fast, and it can thus be used in NRT processing environments. $SO_2$ LH
is retrieved in NRT from TROPOMI UV earthshine spectra in the wavelength range 311-335 nm with an accuracy of better
than 2 km for $SO_2$ columns greater than 20 DU. For low $SO_2$ columns, high-altitude layer heights cannot be retrieved and the
retrieval is biased towards low layer heights (Hedelt et al., 2018). Therefore, the use of the data in the CAMS system is
restricted to values > 20 DU. More details about the retrieval algorithm can be found in Hedelt et al. (2018) and Koukouli et
al. (2021). Koukouli et al. (2021) compared the S5P LH data with IASI observations for the 2019 Raikoke, the 2020
Nishinoshima and the 2021 La Soufrière-St Vincent eruptive periods and found good agreement with a mean difference of
~0.5±3km, while for the 2020 Taal eruption, a larger difference of between 3 and 4±3km was found. In this paper we use
v3.1 of the FP_ILM $SO_2$ LH products.

## 2.3 NRT GOME-2 TCSO₂ data

GOME-2 (Munro et al., 2016) on board the MetOp-A, -B and -C satellites measures in the UV and Vis part of the spectrum
(240-790 nm). MetOp-B and -C have a swath of 1920 km at 40 km x 80 km ground pixel resolution, while MetOp-A has a
narrower swath of 960 km at 40 km x 40 km. Global coverage with GOME-2 is achieved within 1.5 days. The GOME-2
measurements allow for the retrieval of ozone and a range of atmospheric trace gases, including $SO_2$ which is retrieved with
the GOME Data Processor (GDP) developed by DLR and operationally provided by the EUMETSAT's ACSAF that uses a
DOAS method. GDP4.8 is used for GOME-2A and GOME-2B (with a fitting window from 315-326 nm) and GDP4.9 for
GOME-2C (with a fitting window of 312-326 nm to include the strong $SO_2$ line at 313 nm). Input parameters for the DOAS
fit include the absorption cross section of $SO_2$ and the absorption cross sections of interfering gases, ozone and $NO_2$, and a
correction is made in the DOAS fit to account for the ring effect (rotational Raman scattering). An empirical interference
correction is applied to the $SO_2$ slant column values to reduce the interference from ozone absorption (Rix et al., 2012). To
reduce the interference from ozone absorption, the retrieval includes the fitting of two pseudo ozone cross-sections following
the approach of Puķīte et al. (2010). As in the case for the TROPOMI dataset, a volcano activity detection algorithm is used
to identify elevated $SO_2$ values from volcanic eruptions. Such flags were implemented in GDP4.8 (see Table 2) and further
improved in GDP4.9 to use the same flagging as for TROPOMI (see Table 1). CAMS only assimilates the GOME-2 $SO_2$
data that are flagged as volcanic (value=1 for GDP4.8; value=1 or 2 for GDP4.9) and assimilates GOME-2B and GOME-2C
in the NRT system operational in 2021. The GOME-2 data are used at the satellite resolution which is similar to the
resolution of the CAMS model used in this paper. In this paper only $SO_2$ data from GOME-2B are used.

| Flag value | Description |
|---|---|
| 0 | No detection |
| 1 | Elevated $SO_2$ value due to a volcanic $SO_2$ plume |
| 2 | Elevated $SO_2$ value in a region with known increased background level (either anthropogenic pollution or SAA region) |

**Table 2: Volcanic SO₂ flags provided for the GDP4.8 GOME-2A and -2B SO₂ products.**

## 2.4 IASI SO₂ plume altitude data

The Infrared Atmospheric Sounding Instrument (IASI) is flying on board of EUMETSAT's MetOp-A (since 2006), MetOp-B (since 2012) and MetOp-C (since 2017) satellite platforms (Clerbaux et al., 2015). The instruments measure the upwelling radiances in the thermal infrared spectral range extending from 645 to 2760 $cm^{-1}$, with high radiometric quality, 0.5 $cm^{-1}$ spectral resolution. A total of 120 views are collected over a swath of ~ 2200 km using a stare-and-stay mode of 30 arrays of 4 individual elliptical pixels, each of which is 12 km diameter at nadir, increasing at the larger viewing angles. IASI provides global monitoring of total ozone, carbon monoxide, methane, ammonia, nitric acid and $SO_2$, among others atmospheric constituents.

The IASI/MetOp $SO_2$ columnar data are operationally provided by the EUMETSAT's ACSAF. In Clarisse et al. (2012) a novel algorithm for the sounding of volcanic $SO_2$ plumes above ~5 km altitude was presented and applied to IASI observations. The algorithm is able to view a wide variety of total column ranges (from 0.5 to 5000 D.U.), exhibits a low theoretical uncertainty (3–5 %) and near real time applicability which was demonstrated for the recent eruptions of Sarychev in Russia, Kasatochi in Alaska, Grimsvötn in Iceland, Puyehue-Cordon Caulle in Chile and Nabro in Eritrea (Tournigand et al., 2020.) A validation of this algorithm on the Nabro eruption observations using forward trajectories and CALIOP/CALIPSO space-born lidar coincident measurements is presented in Clarisse et al. (2014) where the expansion of the algorithm to also provide $SO_2$ plume altitudes is further described. The IASI/MetOp $SO_2$ ACSAF product includes five $SO_2$ column data at assumed layer heights of 7, 10, 13, 16 and 25 km, as well as a retrieved best estimate for the $SO_2$ plume altitude and associated $SO_2$ column. Note that the $SO_2$ plume altitudes provided by this algorithm are quantized every 0.5km. This dataset is publicly available from https://iasi.aeris-data.fr/SO2_iasi_a_arch/ .

For the requirements of the validation against the CAMS experiments, all available IASI $SO_2$ plume altitude data for the Raikoke volcano 2019 eruption were gridded onto a 1x1° grid at 3h intervals per day. The equivalent CAMS $SO_2$ plume altitude, i.e. the altitude where the maximum $SO_2$ load occurs in the CAMS $SO_2$ profiles, was chosen for the collocations. In the case where two CAMS altitudes provided the same $SO_2$ load, the mean was assigned as the CAMS $SO_2$ plume altitude.

## 3 CAMS global integrated forecasting and data assimilation system

### 3.1 CAMS volcanic SO2 plume forecasting system

The chemical mechanism of ECMWF's Integrated Forecast System (IFS) is a modified and extended version of the Carbon Bond 2005 chemistry scheme (CB05, Yarwood et al. 2005) chemical mechanism for the troposphere, as also implemented in the chemical transport model (CTM) TM5 (Huijnen et al., 2010). CB05 is a tropospheric chemistry scheme with 57 species and 131 reactions. The chemistry module of the IFS is documented in more detail in Flemming et al. (2015) and Flemming et al. (2017) and more recent updates in Inness et al. (2019). The CB05 chemistry scheme is coupled to the AER aerosol bulk scheme (Remy et al. 2019) for the simulation of sulphate, nitrate and ammonium aerosols. More up-to-date information is available from atmosphere.copernicus.eu.

In the original version of the volcanic $SO_2$ plume forecasting system described by Flemming and Inness (2013), there was a dedicated "volcanic $SO_2$ tracer", with oxidation based on a simple fixed timescale approach. By contrast, in the progression of the volcanic $SO_2$ system described here, the volcanic $SO_2$ emissions, and data assimilation of $SO_2$, is applied to the $SO_2$

tracer within the CB05 chemistry scheme (Flemming et al., 2015), with oxidation to sulphate aerosol occurring, based on the
kinetics specified in the chemistry scheme. There are two pathways for this (i) in the gas phase via the hydroxyl radical (OH)
and (ii) within cloud droplets (aqueous phase), with only pathway (i) occurring in the stratosphere (in the model). In the
troposphere, the model includes also the $SO_2$ loss processes of wet deposition and surface dry deposition. Although
heterogenous $SO_2$ oxidation on ash particles, and the self-lofting effect from the ash heating effect, have both been shown to
be important for the $SO_2$ dispersion from Raikoke (Muser et al., 2020) and also from the 2015 Kelud eruption (Zhu et al.,
2020), ash particles are not included in these IFS simulations.

As described in Flemming et al. (2015) the IFS uses a semi-Lagrangian advection scheme. Since the semi-Lagrangian
advection does not formally conserve mass, a global mass fixer is applied to the chemical tracers, including to the $SO_2$ tracer,
and a proportional mass fixer as described in Diamantakis and Flemming (2014) was used for the runs presented in this
paper. More details about the CB05 chemistry scheme can be found in Flemming et al. (2015, 2017), Remy et al. (2018) and
Huijnen et al. (2019).

The model version used in this paper is based on the IFS model cycle 47R1 (CY47R1,
www.ecmwf.int/en/forecasts/documentation-and-support/changes-ecmwf-model), which was the operational CAMS cycle
from 6 October 2020 to 18 May 2021. In CY47R1, the CAMS system uses the CAMS-GLOBANTv4.2 anthropogenic
emissions (Granier et al., 2019) which include anthropogenic $SO_2$, as well as a climatology of $SO_2$ outgassing volcanic
emissions based on satellite data (Carn et al., 2016). Further updates relative to the previous version (CY46R1) are

- change to Global Fire Assimilation System (GFAS) v1.4 biomass-burning emissions
- the exclusion of agricultural waste burning from CAMS_GLOB_ANT to avoid double-counting with GFAS
- improved diurnal cycle and vertical profile for anthropogenic emissions
- introduction of Hybrid Linear Ozone (HLO) scheme, a Cariolle-type linear parameterisation of stratospheric ozone
  chemistry using the multi-year mean of the CAMS reanalysis as mean state
- updated dust source function, which reduces the overestimation of dust in the Sahara, Middle East and other
  regions, and restores missing dust over Australia
- new sea-salt emission scheme based on Albert et al. (2016), which provides better agreement with measured sea-
  salt size distribution
- revised coefficients in UV processor, based on ATLAS3 spectrum.

## 3.2 CAMS data assimilation system

The IFS uses an incremental four-dimensional variational (4D-Var) data assimilation system (Courtier et al. 1994). In the
CAMS 4D-Var a cost function that measures the differences between the model fields and the observations is minimized to
obtain the best possible forecast through the length of the assimilation window by adjusting the initial conditions. $SO_2$ is one
of the atmospheric composition fields that is included in the control vector and minimized together with the meteorological
control variables in the CAMS system (Inness et al., 2015, Flemming and Inness, 2013). The current operational CAMS
configuration uses a weak constraint formulation of 4D-Var which includes a model error term for the meteorological
variables (Laloyaux et al., 2020) that corrects mainly the stratospheric temperature bias and also improves slightly the
stratospheric winds. In the CAMS 4D-Var system, the control variables are the initial conditions at the beginning of the
assimilation window, with the aim of providing the best initial conditions for the subsequent forecast. The background error
covariance matrix in the ECMWF data assimilation system is given in a wavelet formulation (Fisher 2004, 2006). This
allows both spatial and spectral variations of the horizontal and vertical background error covariances. The CAMS

background errors are constant in time. The horizontal resolution of the NRT CAMS 2021 operational system as well as that of the data assimilation experiments presented in this paper is approximately 40 km, corresponding to a triangular truncation of TL511 or a reduced Gaussian grid with a resolution of N256 (more information can be found at https://confluence.ecmwf.int/display/FCST/Gaussian+grids). The operational CAMS system uses two minimisations (the so-called inner loops) at reduced horizontal resolution, currently at TL95 and TL159 corresponding to horizontal resolutions of about 210 km and 125 km. This means that wavenumbers up to 95/159 can be represented in the wavelet formulation for the background errors. For the experiments presented in this paper, slightly higher horizontal resolutions of TL159/TL255 were used for the inner loops (corresponding to about 125 and 80 km, respectively). The CAMS model and data assimilation system has 137 model levels in the vertical, between the surface and 0.01 hPa and uses a 12-hour 4D-Var configuration with assimilation windows from 3 to 15 UTC and from 15-3 UTC.

### 3.2.1 CAMS NRT TCSO$_2$ assimilation configuration (baseline configuration)

The SO$_2$ data assimilated in the CAMS NRT configuration are total column values. To calculate the model equivalent of the observations the CAMS SO$_2$ field is interpolated to the time and location of the measurements and the CAMS SO$_2$ columns are calculated as a simple vertical integral between the surface and the top of the atmosphere. While the background error statistics for most of the atmospheric composition fields (Inness et al., 2015) were either calculated with the National Meteorological Center (NMC) method (Parrish and Derber, 1992) or from an ensemble of forecast differences (following a method described by Fisher and Andersson, 2001), the background errors for SO$_2$ are prescribed by an analytical vertical standard deviation profile and horizontal correlations. SO$_2$ observations are currently only assimilated in the CAMS system in the event of volcanic eruptions. An NMC or ensemble approach would not give useful SO$_2$ background error statistics in these cases as the forecast model does not have information about individual volcanic eruptions, even though it does include emissions from outgassing volcanoes. SO$_2$ background error standard deviations calculated with the NMC or ensemble methods peak near the surface where anthropogenic SO$_2$ concentrations are largest and will hence lead to the largest analysis increments near the surface. Therefore, for the assimilation of volcanic SO$_2$ data, background error statistics for SO$_2$ were constructed by prescribing a background error standard deviation profile that is a delta function and peaks in the mid troposphere around model level 98 (about 550 hPa) in the 137 level model version, corresponding to an SO$_2$ plume height of about 5 km (see blue profile in Figure 1).

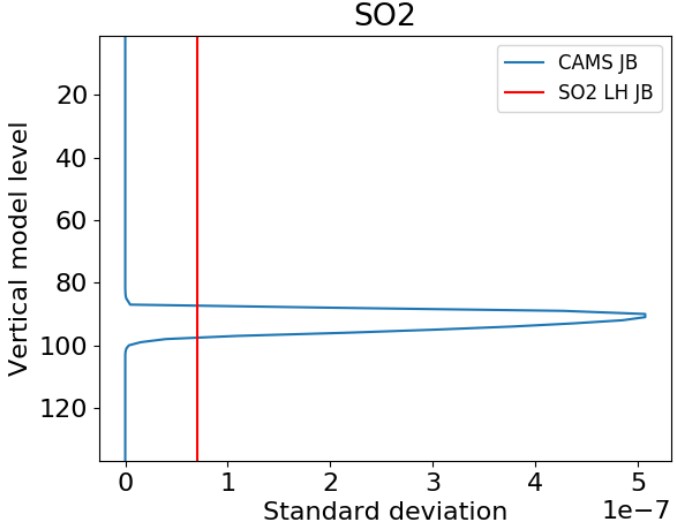

**Figure 1: Vertical profile of SO$_2$ background error standard deviation in kg/kg used in the operational CAMS configuration (blue) and for the main LH experiment (red, LHexp). The y-axis shows model levels. Level 1 is the top of the atmosphere, level 137 the surface.**

The SO$_2$ wavelet file in the NRT CAMS configuration (also called baseline configuration in this paper) is formed of diagonal vertical wavenumber correlation matrices, with the value on the diagonal controlled by a horizontal Gaussian correlation function with a standard deviation of 250 km and a globally constant vertical standard deviation profile. The values of the elements on the diagonal of the vertical correlation matrix are the same at every level but vary for each wavenumber. If TCSO$_2$ data are assimilated the largest correction to the model's background will be applied where the background errors are largest, i.e. in the mid-troposphere around 550 hPa. The CAMS SO$_2$ analysis is univariate, i.e. there are no cross correlations between SO$_2$ background errors and the other atmospheric composition control variables.

### 3.2.2 Data assimilation configuration for TCSO$_2$ LH data

If information about the altitude of the volcanic SO$_2$ layer is known in NRT a different approach can be followed. In this case, we use a background error standard deviation profile that is constant in height (e.g. red line in Fig. 1) and calculate the SO$_2$ column not between the surface and the top of the atmosphere, but between the pressure values that correspond to the bottom and the top of the retrieved volcanic SO$_2$ layer. The depth of this layer is currently set in the FP_ILM product as 2 km, which corresponds to the uncertainty of the retrieved layer height. This approach mimics the procedure of using TROPOMI SO$_2$ averaging kernels which are box profiles, but for the retrieved layer and not an assumed hypothetical volcanic SO$_2$ profile (see TROPOMI SO$_2$ ATBD, http://www.tropomi.eu/documents/). One limitation of this method is that the SO$_2$ LH product gives the plume altitude with an accuracy of 2 km, but does not give a value for the lower vertical boundary of the SO$_2$ plume, and for a thick plume part of the SO$_2$ loading could be missed in the calculation of the model equivalent. However, as the model's background SO$_2$ concentrations in the free troposphere are low this should not be a big issue in the column calculation. Also, some vertical variation of the SO$_2$ loading will result from assimilating observations with varying plume altitudes for larger volcanic plumes that are not uniform in height everywhere, and Figure 3 below shows that this is indeed the case for the Raikoke eruption. Results from sensitivity studies regarding the choice of the constant background error standard deviation value are given below in Section 4.2.

## 4. Assimilation of TROPOMI TCSO$_2$ data for 2019 Raikoke eruption

### 4.1 Raikoke eruption June 2019

The Raikoke volcano, located on the Kuril Islands south of the Kamchatka peninsula, erupted around 18 UTC on 21 June 2019 and emitted SO$_2$ and ash in a series of explosive events until about 6 UTC on 22 July. The SO$_2$ and ash plume rose to around 8-18 km (Muser et al., 2020; Grebennikov et al., 2020) meaning a considerable amount of the SO$_2$ reached the stratosphere. The volcanic cloud was transported around much of the northern hemisphere, was observed by TROPOMI and GOME-2 for about a month and was also observed with ground-based measurements (Vaughan et al., 2021; Grebennikov et al., 2020) and other satellites (Muser et al., 2020). Figure 2 shows the TCSO$_2$ burden from the Raikoke eruption as calculated from NRT TROPOMI and GOME-2B data. All the TCSO2 satellite data available during a 12-hour assimilation window were gridded onto a 1°x1° degree grid and the area of all grid cells with SO$_2$ values greater than the listed threshold values was calculated. For a threshold of 1 DU the SO$_2$ burdens from TROPOMI and GOME-2B were around 1.5 Tg and 1.1 Tg, respectively. These values agree with findings by de Leeuw et al. (2021) and make the eruption the largest since the eruption of the Nabro volcano in 2011 (de Leeuw et al, 2021; Goitom et al, 2015; Clarisse et al., 2014). The 'dip' in the TROPOMI SO$_2$ burden after the initial peak is an artefact that results from missing observations in the TROPOMI NRT data on 25 June 2019 in the area of highest SO$_2$ values (also visible in Figure 9c2 below).

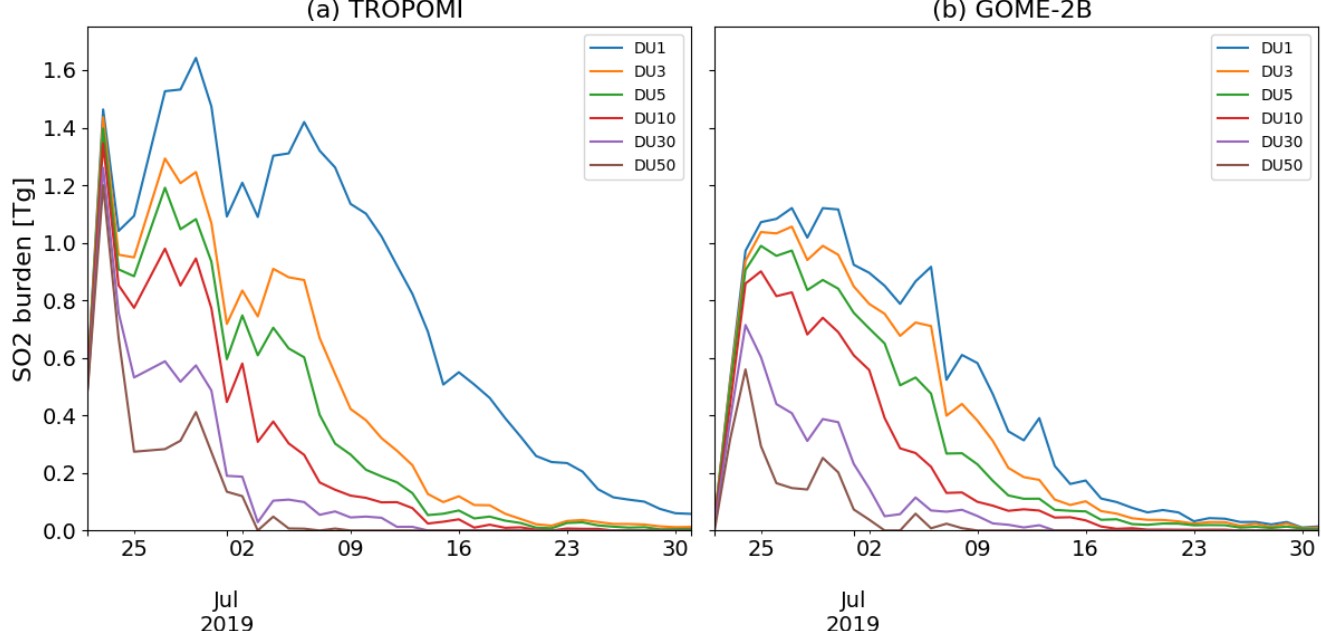

**Figure 2: SO₂ burden (in Tg) from TROPOMI (left) and GOME-2B (right) from 22 June to 31 July 2031. The values are calculated by gridding the data on a 1°x1° grid and selecting the grid cells with TCSO₂ values greater than thresholds of 1, 3, 5, 10, 30 and 50 DU in the area 30-90°N.**

Figure 3 shows a timeseries of the SO₂ LH information from the TROPOMI LH product for the Raikoke plume. It shows that volcanic SO₂ can be detected and the SO₂ LH information retrieved for about 3 weeks after the eruption. The bulk of the SO₂ was located above 300 hPa, (about 9 km) with a considerable amount above 200 hPa (about 12 km). This is considerably higher than the 550 hPa that is assumed as the plume location in the CAMS operational (baseline) configuration. Large TCSO₂ values (>100 DU) were observed in the first days after the eruption.

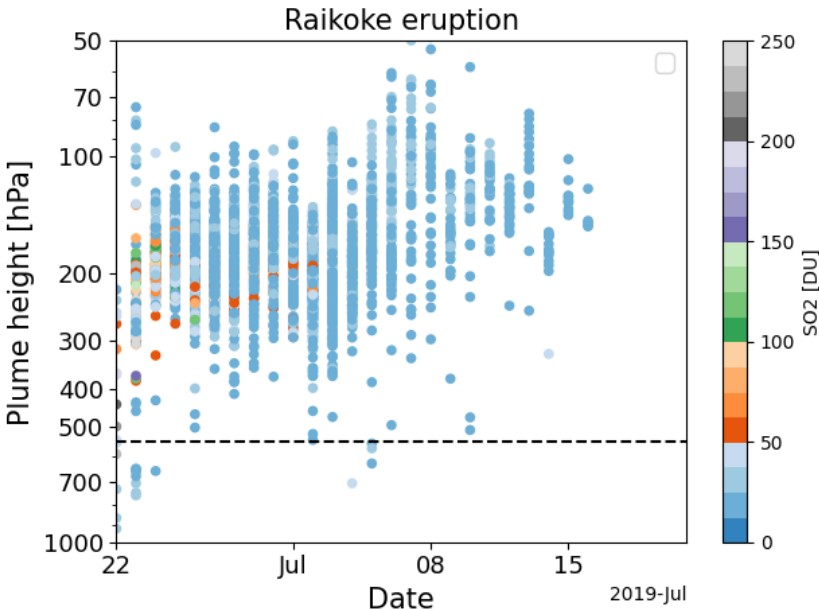

**Figure 3: Timeseries of the height of the Raikoke volcanic plume (averaged over 30-90°N) in hPa from TROPOMI SO₂ LH data from 22 June to 21 July 2019. The colours show the corresponding TCSO₂ values in DU. The dashed horizontal line at 550 hPa shows the altitude where the CAMS baseline configuration places the maximum SO₂ increment.**

**4.2 Sensitivity studies for assimilation of TCSO₂ data**

Several data assimilation experiments were run for the period 22 June to 21 July 2019 to test the assimilation of the SO₂ LH
data and to compare the results with the CAMS baseline configuration, listed in Table 3. The baseline experiment (BLexp)

| Experiment Abbreviation | Experiment ID, DOI | Assimilated SO₂ data | Bg-error stdv [kg/kg] | Bg-error hcor [km] | Resolution of minimisations |
|---|---|---|---|---|---|
| BLexp | hhu5, [10.21957/cygt-xf49](10.21957/cygt-xf49) | S5P NRT > 5DU | CAMS (see Fig. 2) | 250 | TL159, TL255 |
| LHexp | hgze, [10.21957/qfam-7474](10.21957/qfam-7474) | S5P LH >20DU | $0.7e^{-7}$ | 100 | TL159, TL255 |
| LH50 | hhbu, [10.21957/zpdt-f079](10.21957/zpdt-f079) | S5P LH > 20DU | $1e^{-7}$ | 50 | TL159, TL255 |
| LH100 | hhtm, [10.21957/jraa-s174](10.21957/jraa-s174) | S5P LH> 20DU | $1e^{-7}$ | 100 | TL159, TL255 |
| LH250 | hhtn, [10.21957/ddxs-2v95](10.21957/ddxs-2v95) | S5P LH > 20DU | $1e^{-7}$ | 250 | TL159, TL255 |
| LH1.4 | hgz7, [10.21957/81bh-7h58](10.21957/81bh-7h58) | S5P LH > 20DU | $1.4e^{-7}$ | 100 | TL159, TL255 |

**Table 3: List of SO₂ assimilation experiments used in this paper. The main experiments discussed in Section 4 are the baseline experiment (BLexp) and the layer height experiment (LHexp). The additional experiments are used in the sensitivity studies in Section 4.2. Bg-error denotes background error, stdv standard deviation and hcor horizontal correlation length scale.**

which assimilated NRT TROPOMI TCSO₂ data with the operational CAMS configuration and the layer height experiment
(LHexp) which uses the FP_ILM S5P LH data with a horizontal background error correlation length of 100 km and
background error standard deviation values of $0.7e^{-7}$ kg/kg are the main experiments used in this paper (Section 4.3 below) to
assess if the assimilation of the SO₂ LH data using a more realistic height rather than the default 5 km improves the CAMS
SO₂ analyses and forecasts. The other LH experiments assess the impact of using different horizontal SO₂ background error
correlation length scales and various SO₂ background error standard deviation values. In all these experiments GOME-2 SO₂
data were not assimilated, and GOME-2B is used as a fully independent dataset for the validation.

The low resolution of the minimisation (TL95/TL159 in the CAMS system operational in 2021) is a factor that limits the
ability of the SO₂ analysis to reproduce small-scale SO₂ features seen in the observations because it gives a lower limit for
the length scale of the horizontal background error correlations that can be used, i.e. for the operational CAMS configuration
only wavenumbers up to 95/159 can be represented. The smallest wavelength ($\lambda_{min}$) that can be represented by two grid
points on a linear grid is

$$\lambda_{min} = \frac{2\pi R}{n_{max}} \quad (1)$$

where R is the radius of the Earth and $n_{max}$ the maximum wavenumber of the truncation (95 or 159 for the inner loops in the
operational CAMS configuration), i.e. twice the size of a grid box. This means that the minimum wavelengths which can be
represented with two grid points for TL95, TL159 and TL255 are about 420 km, 250 km, 160 km, respectively and smaller
scale horizontal structures cannot be represented in the background error wavelet formulation. Figure 4 illustrates this and
shows horizontal SO₂ correlations at the surface for horizontal background error length scales of 50km, 100km and 250 km
for truncations of TL95, TL159 and TL255. The 'wriggles' seen in the TL95 (and to a lesser extent in the TL159) plots show
that the shorter background error correlations length scales cannot be properly resolved at these truncations. Even at TL255
some minor oscillations are still visible for horizontal correlation length scales of 50 km. Therefore, to properly resolve

smaller-scale plumes the resolutions of the inner loops would need to be even higher than TL255. Figure 4 also illustrates how far an increment from a single SO$_2$ observation would be spread out in the horizontal and therefore affect grid points away from the observation.

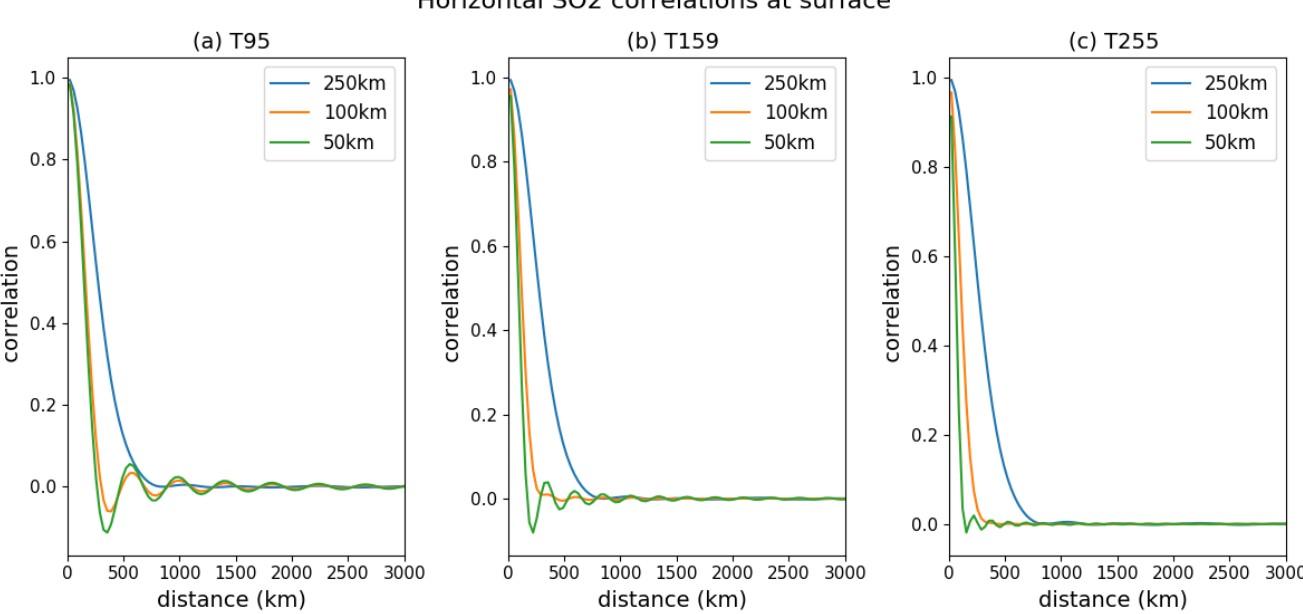

**Figure 4: SO$_2$ background error horizontal surface correlations at different truncations: (a) TL95, (b) TL159 and (c) TL255 if the horizontal length scales are specified as Gaussian correlation function with length scales of 250 km (blue), 100 km (orange) and 50 km (green).**

The operational NRT CAMS configuration uses minimisations at TL95/TL159 and a length scale of 250 km for the horizontal SO$_2$ background error correlations. For the data assimilation experiments shown in this paper we use inner loops of TL159/TL255 to allow us to use a Gaussian correlation function with a length scale of 100 km and therefore resolve slightly smaller-scale features than in the operational NRT CAMS system. The computational cost of one analysis cycle increases by about 20-30% when the spectral resolution of the minimisation is increased in this way, with the largest increase coming from the second minimisation which is about 50% computationally more expensive than at lower resolution. Figure 5 shows the CAMS TCSO$_2$ analysis fields on 27 June 2019 resulting from the assimilation of the TROPOMI SO$_2$ LH data when horizontal background error correlation length scales of 50, 100 and 250 km were used (experiments LH50, LH100, LH250), while using the same background error standard deviation profile of 1e$^{-7}$ kg/kg in all cases. Also shown are the NRT TROPOMI and GOME-2B TCSO$_2$ data for that day. The figure illustrates the large impact of the horizontal background error correlation length scale on the SO$_2$ analysis, as the SO$_2$ plume is considerably more spread out in the CAMS analysis when longer horizontal correlations are used, and that better agreement with the features seen in the observations is found for shorter horizontal correlations. Figure 6 shows timeseries of SO$_2$ burden and plume area for a threshold of 5 DU from TROPOMI, GOME-2B and the three SO$_2$ LH experiments to further assess the impact on the SO$_2$ analysis of changing the horizontal correlation

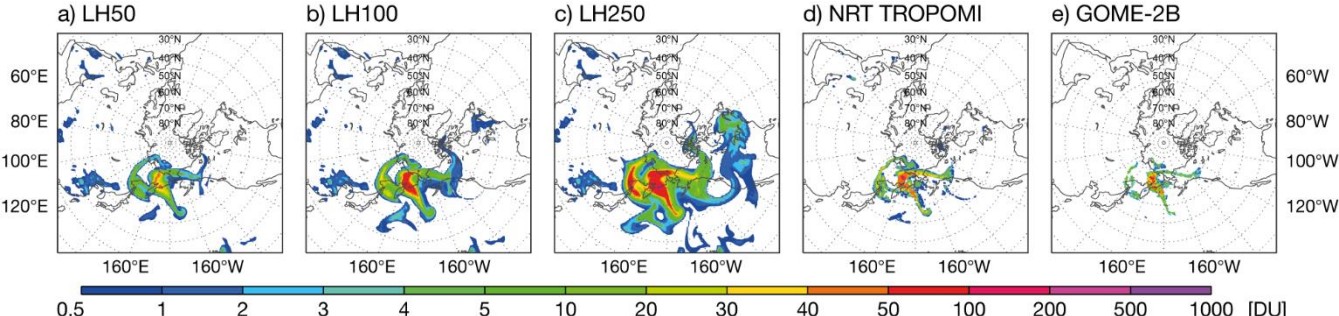

**Figure 5: TCSO₂ analyses on 27 June 2019 at 0z obtained by assimilating SO₂ LH data using a background standard deviation profile of 10⁻⁷ kg/kg and background errors with horizontal correlations of (a) 50 km, (b) 100 km and (c) 250 km. Also shown are (d) NRT TROPOMI and (e) GOME-2B TCSO₂ values.**

length scale. We see that the SO₂ burden and plume area calculated from the observations are overestimated by all three CAMS TCSO₂ analyses. This overestimation is a well-known feature usually seen in the operational NRT CAMS volcanic SO₂ assimilation. Figure 6 illustrates that a major factor causing this overestimation is the choice of the horizontal background error correlation length scale and that by choosing a length scale of 250 km the SO₂ burden and plume area are about 6 times larger than for a length scale of 50 km. This implies that a limiting factor for correctly reproducing the SO₂

burden and plume area in the CAMS analysis is the resolution of the inner loops as it limits the horizontal correlation length scale that can be chosen for the background errors. A coarser inner loop resolution requires a longer horizontal length scale because shorter wavelengths cannot be resolved properly. If the aim is to reproduce finer-scale volcanic plumes with the CAMS data assimilation system, the horizontal resolution of the inner loops will have to be increased. For the main LH experiment used in this paper we decided to use a horizontal correlation length scale of 100 km which can be represented

properly if the resolutions of the inner loops are TL159/TL255.

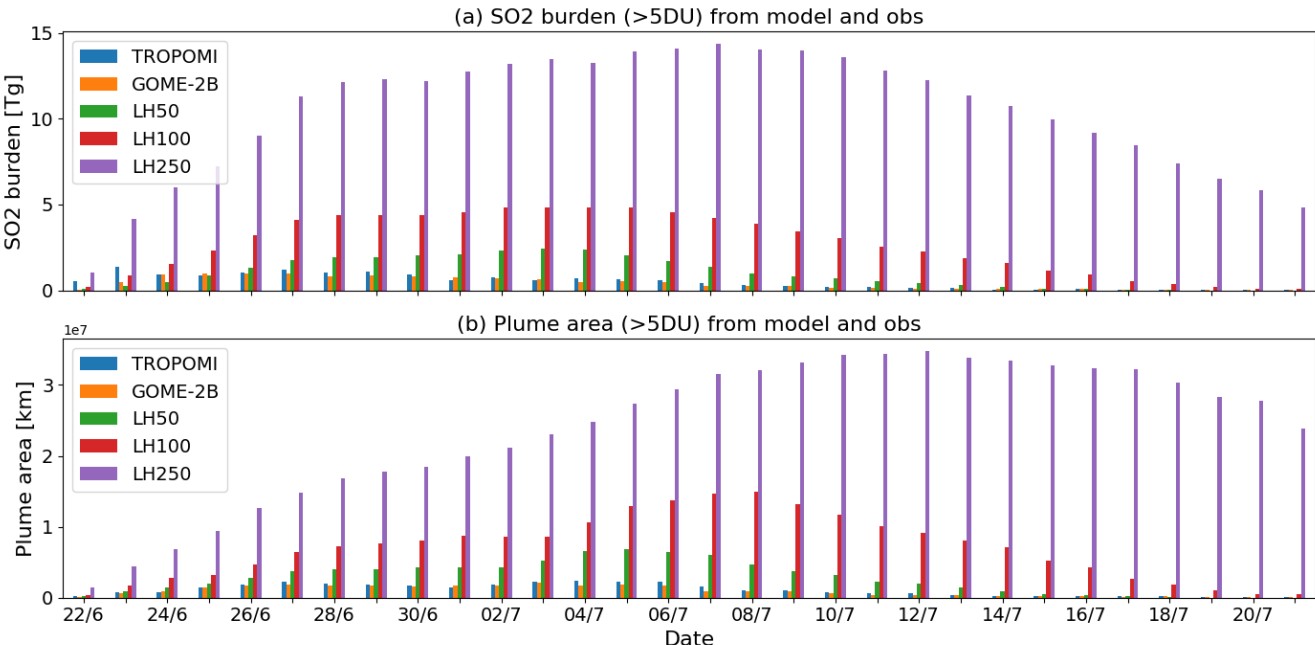

**Figure 6: (a) SO₂ burden in Tg and (b) plume area in 1e⁷ km² from TROPOMI, GOME-2B and three SO₂ LH experiments at 0z with horizontal background error length scales of 50 km (LH50), 100km (LH100) and 250 km (LH250) for the Raikoke eruption (22 June to 21 July 2019). The values are calculated by gridding the data on a 1°x1° grid and selecting the grid cells with TCSO₂**

**values greater than 5 DU in the area 30-90°N.**

Another factor that influences the results of the SO₂ analysis is the value of the background error standard deviation profile. This is illustrated in Figure 7 which shows time series of SO₂ burden and plume area from TROPOMI, GOME-2B and three SO₂ LH experiments with varying background error standard deviation values (0.7e⁻⁷, 1.0e⁻⁷, 1.4e⁻⁷kg/kg). All experiments

used a horizontal background error correlation length scale of 100 km. The larger the background error standard deviation,

the larger the correction that is made by the SO$_2$ analysis and the larger the SO$_2$ burden and plume area become. However, the impact of changing the background error standard deviation is not as big as changing the horizontal background error correlation length scale and increasing the standard deviation value from 0.7e$^{-7}$ kg/kg to 1.4e$^{-7}$ kg/kg doubles the SO$_2$ burden and plume area.

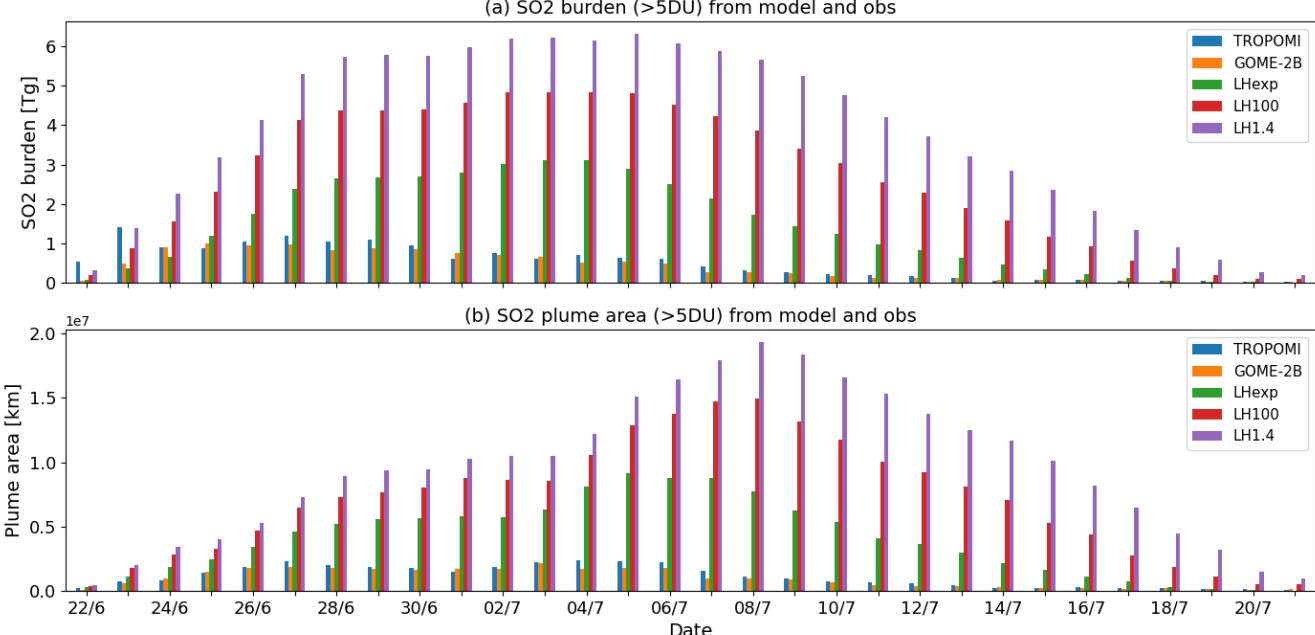


**Figure 7: (a) SO$_2$ burden in Tg and (b) plume area in 1e$^7$ km$^2$ from TROPOMI, GOME-2B and three SO$_2$ LH experiments at 0z with background error standard deviation values of 0.7e$^{-7}$ (LHexp), 1e$^{-7}$ (LH100) and 1.4e$^{-7}$ kg/kg (LH1.4) for the Raikoke eruption (22 June to 21 July 2019). The values are calculated by gridding the data on a 1ºx1º grid and selecting the grid cells with TCSO$_2$ values greater than 5 DU in the area 30-90ºN.**


For the remainder of this paper the LH experiment that uses a value of 0.7e-7 kg/kg for the background error standard deviation and a horizontal background error correlation length scale of 100 km is used (abbreviated as LHexp).

### 4.3 Results of TCSO$_2$ assimilation tests for the Raikoke 2019 eruption

The SO$_2$ analysis fields and 5-day forecasts for the Raikoke eruption from the SO$_2$ layer height experiment (LHexp) and the
baseline experiment with the CAMS configuration (BLexp) are now assessed in more detail. This assessment includes (1) a visual inspection of the SO$_2$ analysis, (2) the assessment of the vertical location of the analysis SO$_2$ plume by comparison with independent IASI/ MetOp plume height observations and (3) the assessment of the quality of the 5-day SO$_2$ forecasts that are started from the LHexp and BLexp SO$_2$ analyses.

We evaluate the SO$_2$ analyses and forecasts against GOME-2B and TROPOMI NRT TCSO$_2$ data. GOME-2B TCSO$_2$ data are fully independent because they are not used in our SO$_2$ assimilation experiments, and TROPOMI NRT TCSO$_2$ products are useful to demonstrate in how far the analyses manage to reproduce the TROPOMI TCSO$_2$ values. It has to be kept in mind that the version of the SO$_2$ LH product used in this study (v3.1) attains its optimal accuracy of 2km for SO$_2$ columns greater than 20 DU and hence, in LHexp, no TCSO$_2$ observations below 20 DU are assimilated. For the evaluation, the SO$_2$
analyses and forecasts, as well as the satellite data, are gridded onto a 1ºx1º grid. Figure 8 shows a timeseries of the number of observations that are actively assimilated in both experiments, i.e. the number of 1ºx1º grid points with active observations, and illustrates that there are more active data in BLexp where NRT TROPOMI SO$_2$ data with values greater than 5 DU are assimilated (i.e. as done in the operational CAMS system) than in LHexp where only data with LH TCSO$_2$ greater than 20 DU are assimilated.

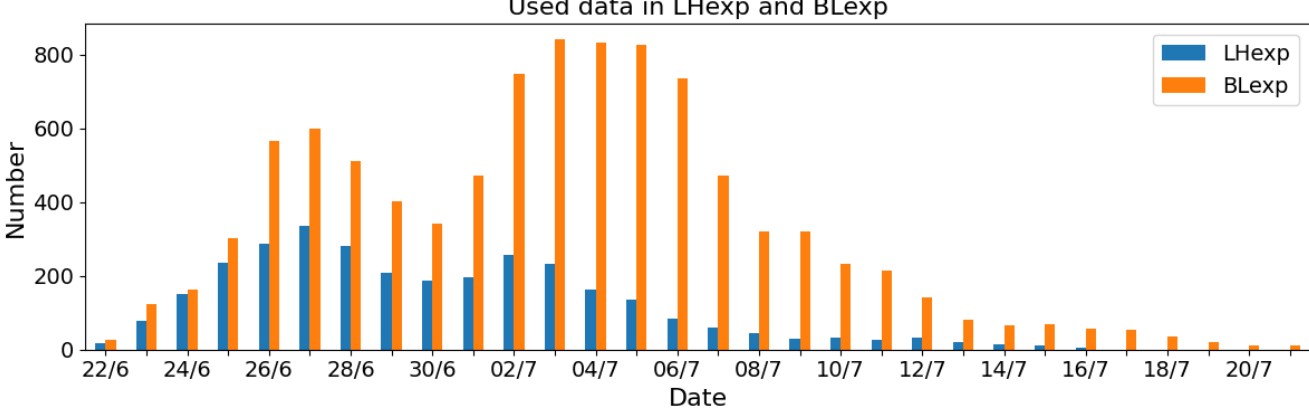


**Figure 8: Timeseries of number of active TROPOMI SO₂ observations assimilated in LHexp (blue) and BLexp (orange) both gridded on a 1°x1° grid (22 June to 21 July 2019).**

### 4.3.1 Evaluation of TCSO₂ analyses

Figure 9 shows TCSO₂ maps from LHexp and BLexp as well as maps of TCSO₂ from NRT TROPOMI, GOME-2B and FP_ILM TROPOMI SO2LH data for 4 days: 22, 25, 29 June and 4 July 2019. The maps on 22 June capture the beginning of the eruption and show that the TCSO₂ values from the first analysis cycle in both experiments are lower than the observations. It also illustrates that even at this initial time the extent of the SO₂ plume is overestimated in both experiments. By 25 and 29 June the SO₂ plume already covers a big part of the North Pacific and by 4 July SO₂ from the eruption is

detected in half the northern hemisphere. LHexp captures the structures of the SO₂ plumes seen in the observations better than BLexp, but overall, both experiments capture the horizontal extent of the plume reasonably well. Figure 9 also illustrates that GOME-2B and NRT TROPOMI TCSO₂ show the same features of the plume, however the TROPOMI NRT lower detection limit facilitates the retrieval of smaller TCSO₂ values around the edges of the plumes. The FP_ILM SO2LH product (v3.1) does not provide reliable information for TCSO₂ < 20 DU and therefore only picks up those parts of the

plume that are associated with the highest SO₂ load. This also explains the lower number of active observations seen in Fig.8. Especially during the later stages of the eruption parts of the plume are missed by the SO₂ LH product. Nevertheless, when assimilating the FP_ILM SO₂ LH data we find good agreement with the NRT TROPOMI data and the GOME-2B data in LHexp (Fig. 9, column 1) when the CAMS analysis reports SO₂ values < 20DU.

Figure 10 shows timeseries of the SO₂ burden from NRT TROPOMI, GOME-2B and the two experiments calculated for threshold values of 5 DU and 30 DU, and Figure 11 shows the corresponding timeseries of the plume areas. For the lower

threshold of 5 DU both the SO₂ burden and the plume area are overestimated in LHexp and BLexp. This confirms what was

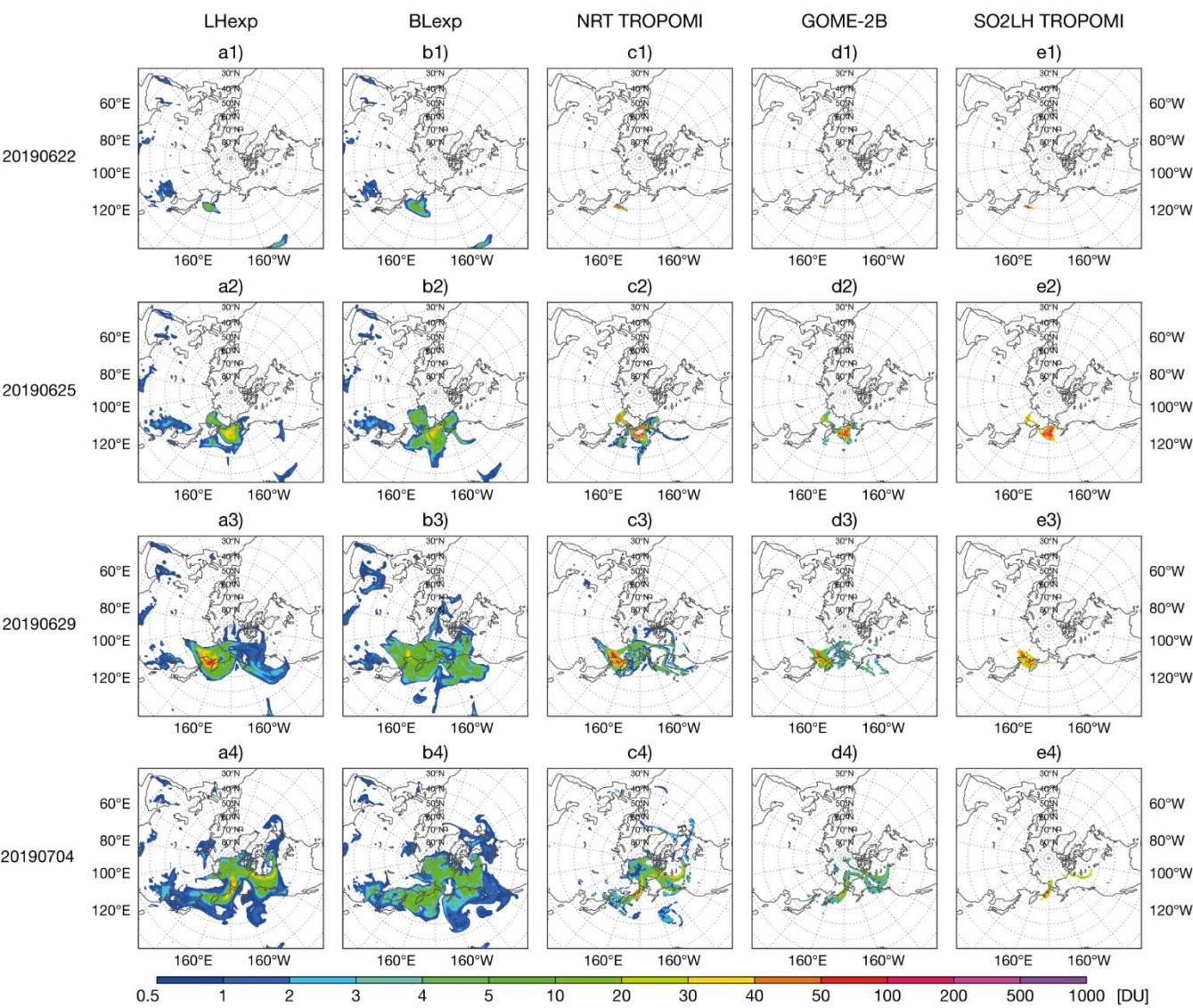

**Figure 9: TCSO₂ analysis fields at 0z from LH exp (a), BL experiment (b), NRT TROPOI (c), NRT GOME-2B (d) and TROPOMI SO₂LH (e) on 22 June (row 1), 25 June (row 2), 29 June (row 3) and 4 July (row 4) in DU. In panels (c)-(e) all available observations are shown, illustrating that the SO₂ LH product only picks up those parts of the plume that are associated with the highest SO₂ load.**

already seen in Figures 6 to 8, namely that the plumes are more spatially dispersed in the analysis than in the observations. The overestimation of the SO₂ burden is larger in LHexp than in BLexp with maximum values of 3 Tg and 2 Tg, respectively, compared to 1.5 and 1.2 Tg for NRT TROPOMI and GOME-2B. However, the plume area is larger in BLexp with maximum extent of about $1e^7$ km², compared to $0.8e^7$ km² in LHexp and $0.2e^7$km² calculated from the observations. The larger overestimation of the SO₂ burden in LHexp is the result of differences in the background error standard deviation values used in the experiments and of the fact that lower SO₂ columns, which could correct an overestimation in parts of the plume, are not assimilated. BLexp fails to capture the higher SO₂ column values, leading to an underestimation of plume area and SO₂ burden for a threshold of 30 DU, while LHexp does have TCSO₂ values > 30 DU but overestimates both plume area and SO₂ burden.

To quantify the realism of the SO₂ analyses and the quality of the SO₂ forecasts appropriate error measures need to be defined and used in addition to the visual inspection of the SO₂ plumes. Statistical measures such as bias and root mean square error are not well suited because of the specific event character of the SO₂ plumes. In addition to looking at the plume

area and SO₂ burden, we use threshold-based measures based on the number of hits (grid boxes where both model and observations detect the plume), misses (grid boxes where there is a plume in the observations but not in the model) or false alarms (grid boxes where the model has volcanic $SO_2$ that is not seen in the observations) to quantify the error in the plume

position. In Flemming and Inness (2013) we used hits and plume area measures for various thresholds. In this paper we combine the information about hits and misses and use as score the probability of detection (POD)

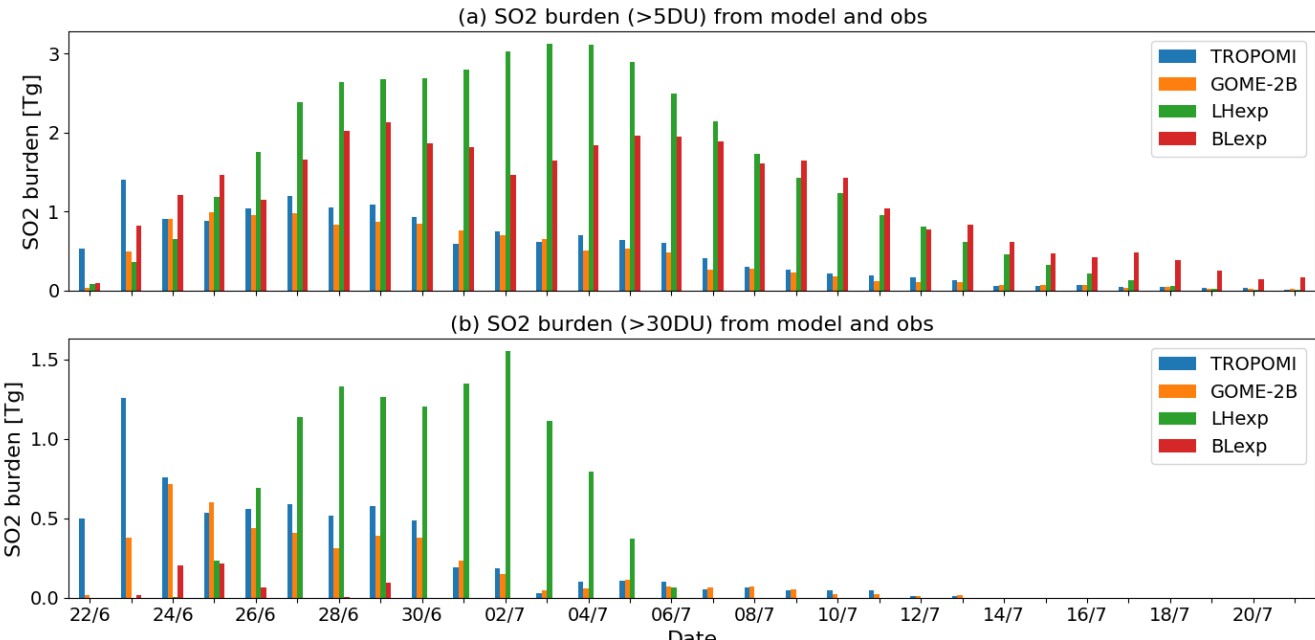

**Figure 10: SO₂ burden in Tg from TROPOMI, GOME-2B, LHexp and BLexp TCSO₂ analysis at 0z for the Raikoke eruption (22 June to 21 July 2019). The values are calculated by gridding the data on a 1⁰x1⁰ grid and selecting the grid cells with TCSO₂ values**
**greater than (a) 5 DU and (b) 30 DU in the area 30-90⁰N.**

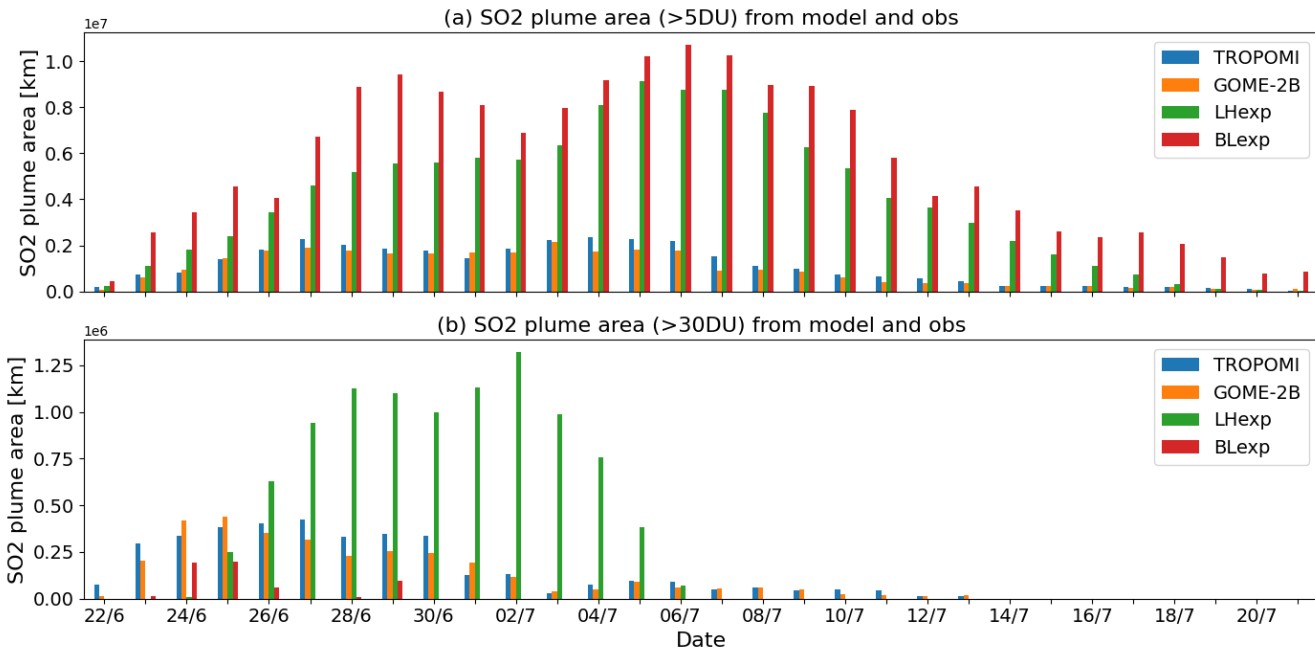

**Figure 11: SO₂ plume area [km²] from TROPOMI, GOME-2B, LHexp and BLexp TCSO₂ analysis at 0z for the Raikoke eruption (22 June to 21 July 2019). The values are calculated by gridding the data on a 1⁰x1⁰ grid and selecting the grid cells with TCSO₂ values greater than (a) 5 DU and (b) 30 DU in the area 30-90⁰N.**


$$POD = hits/(hits+misses) \quad (2)$$

which lies between 0 and 1, with a value of 1 indicating a perfect score. We also us the critical success index (CSI), defined as

$$CSI=hits/(hits+misses+false\ alarms)\quad(3)$$

which additionally considers the number of false alarms and again has values between 0 and 1 with 1 indicating a perfect score (Nurmi, 2003). These are point based comparisons and might score badly for features that are close but slightly misplaced between observations and model.

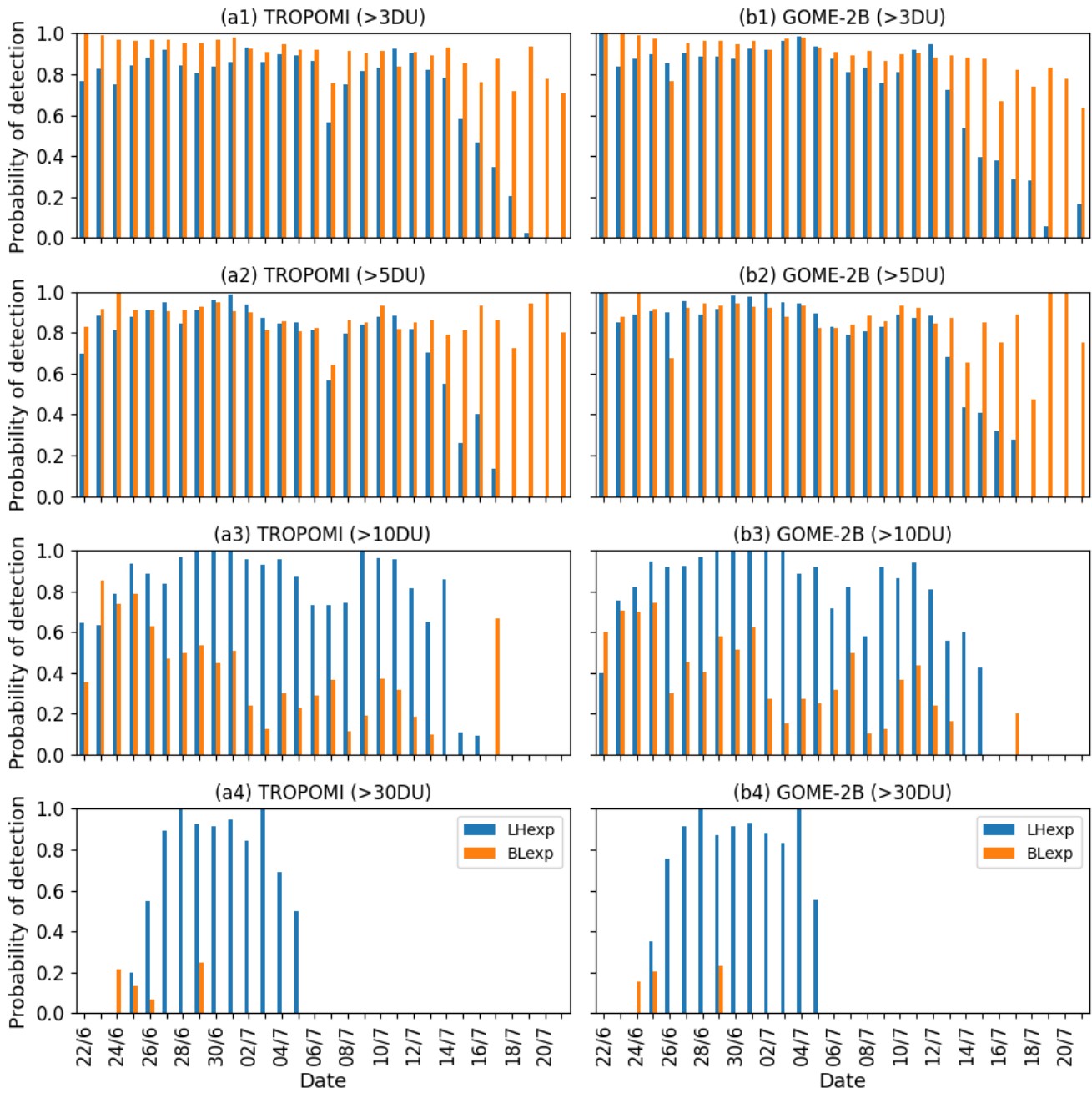


**Figure 12: Timeseries of POD for TCSO₂ analysis fields (at 0z) against (a) NRT TROPOMI and (b) GOME-2B for TCSO₂ thresholds of (1) 3DU, (2) 5 DU, (3) 10 DU and (4) 30 DU (22 June to 21 July 2019). Values for LHexp are shown in blue, values for BLexp in orange.**

Figure 12 shows the POD from LHexp and BLexp for various TCSO₂ analysis thresholds (3, 5, 10, 30 DU) scored against NRT TROPOMI and GOME-2B data. The results are very similar for both satellites. The parts of the plume with lower TCSO₂ values are well captured by both experiments with POD values above 0.9 for BLexp for most of the period and POD

values above 0.8 for LHexp. The POD in LHexp decreases towards the end of the depicted period because the number of assimilated data drops strongly (see Fig. 8), while more observations are assimilated in BLexp at the later stage of the

episode. BLexp, however, does not capture the higher values observed by NRT TROPOMI and GOME-2B well while LHexp has a much higher POD for those parts of the plume. No values above 30 DU are detected after 5 July 2019.

Figure 13 shows the CSI from LHexp and BLexp, the measure that also penalises the false alarms. As expected, these values are considerably lower than the POD (with maximum values around 0.6) because plume area and $SO_2$ burden are

overestimated in both experiments (see Fig. 9) leading to numerous false alarms. Both experiments behave similarly for the lower thresholds but $TCSO_2$ values greater than 30 DU are again captured better in LHexp.

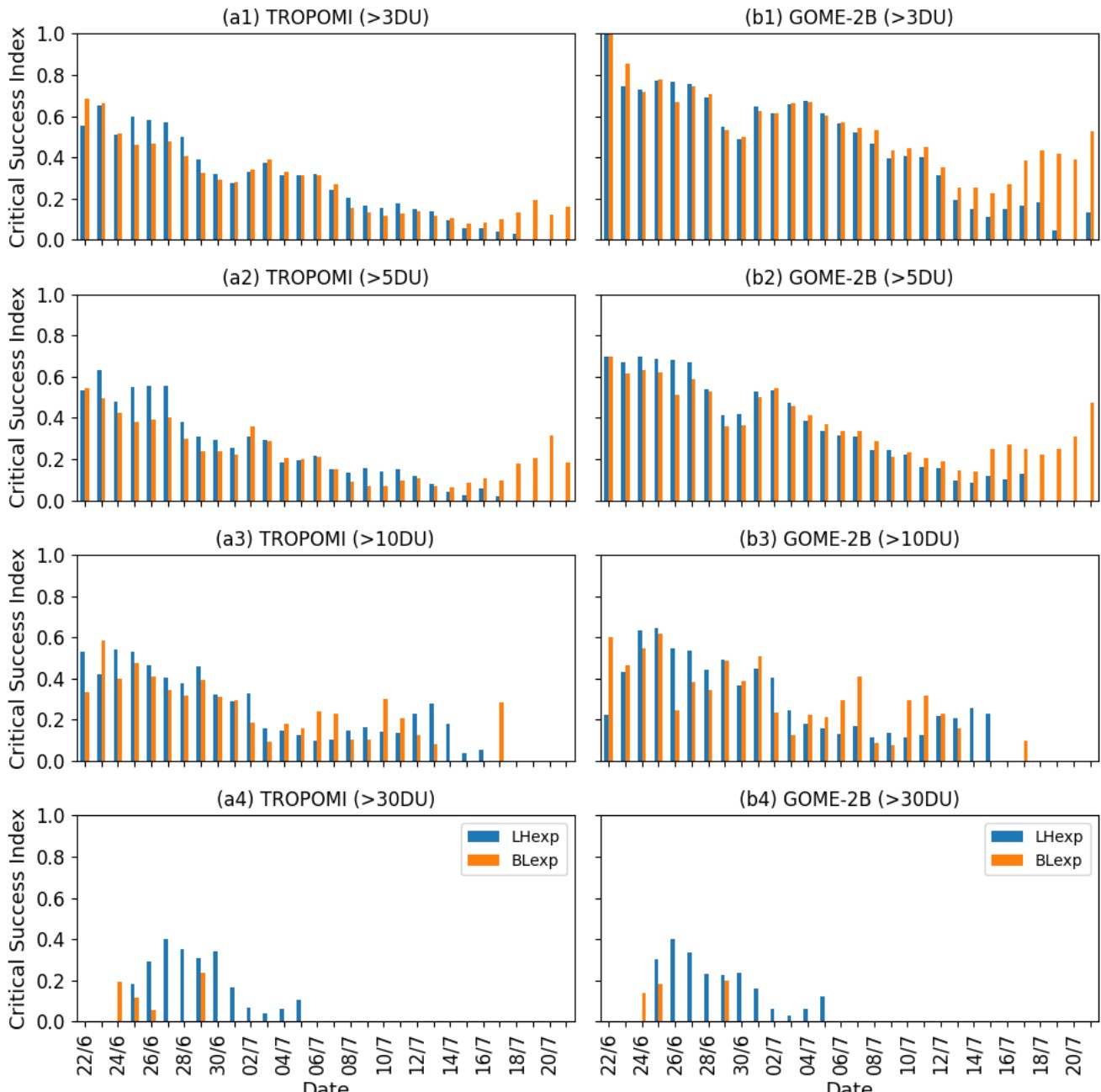

**Figure 13: Timeseries of CSI for TCSO₂ analysis fields (at 0z) against (a) NRT TROPOMI and (b) GOME-2B for TCSO₂ thresholds of (1) 3DU, (2) 5 DU, (3) 10 DU and (4) 30 DU (22 June to 21 July 2019). Values for LHexp are shown in blue, values for**
**BLexp in orange.**

In summary, as far as the TCSO$_2$ analysis fields are concerned the performance of LHexp and BLexp is similar for TCSO$_2$ columns below 10 DU, but BLexp does not capture the higher SO$_2$ values as well as LHexp. Both experiments overestimate the SO$_2$ burden and the plume area compared to the TROPOMI NRT and GOME-2B observations.


### 4.3.2 Vertical location of the SO$_2$ plume

While the TCSO$_2$ analyses from LHexp and BLexp score similarly in the detection of the TCSO$_2$ plume observations by GOME-2B and NRT TROPOMI, at least for values less than 10 DU, the vertical distributions of the SO$_2$ plumes from the experiments differ considerably. Figure 14 shows vertical cross sections along 60°N between 120-300°E through the SO$_2$

plume on 29 June 2019 from LHexp and BLexp. The figure illustrates that the bulk of the SO$_2$ plume is located between 200-100 hPa in LHexp while it is located much lower, between 600-400 hPa, in BLexp. To assess which vertical distribution is more realistic, in Figure 15 we compare the plume heights from the experiments with SO$_2$ altitudes derived from IASI LATMOS ULB data (Clarisse et al., 2012) for the period 22 to 29 June 2019. The CAMS plume altitude was calculated as the altitude where the highest SO$_2$ value were found in the CAMS SO$_2$ profiles. The figure shows that the plume height in

LHexp agrees well with the independent IASI plume altitude with a mean bias of 0.4±2.2 km, while BLexp underestimates the plume altitude with a mean bias of -5.1±2.1 km. Figure 15 illustrates that the altitude of the Raikoke SO$_2$ plume in the CAMS analysis is considerably improved if SO$_2$LH data are used than when using the baseline configuration.

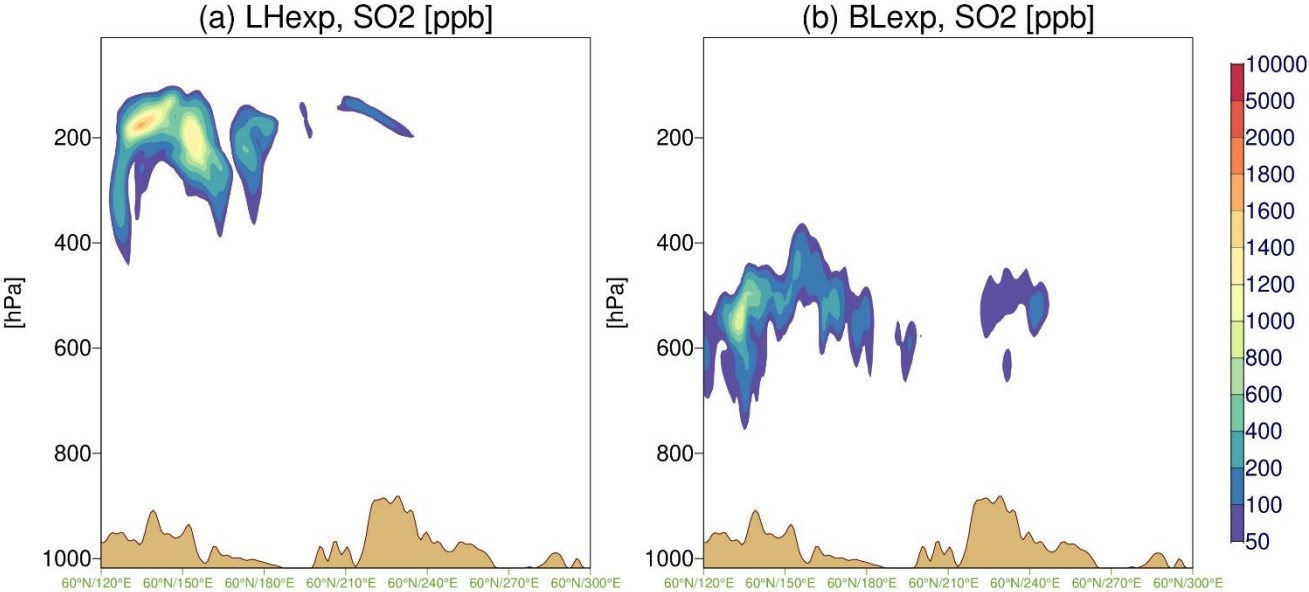

**Figure 14: Vertical cross sections along 60°N between 120-300°E showing the SO$_2$ analysis field (in ppb) from (a) LHexp and (b)**
**BLexp on 29 June 2019, 0z.**

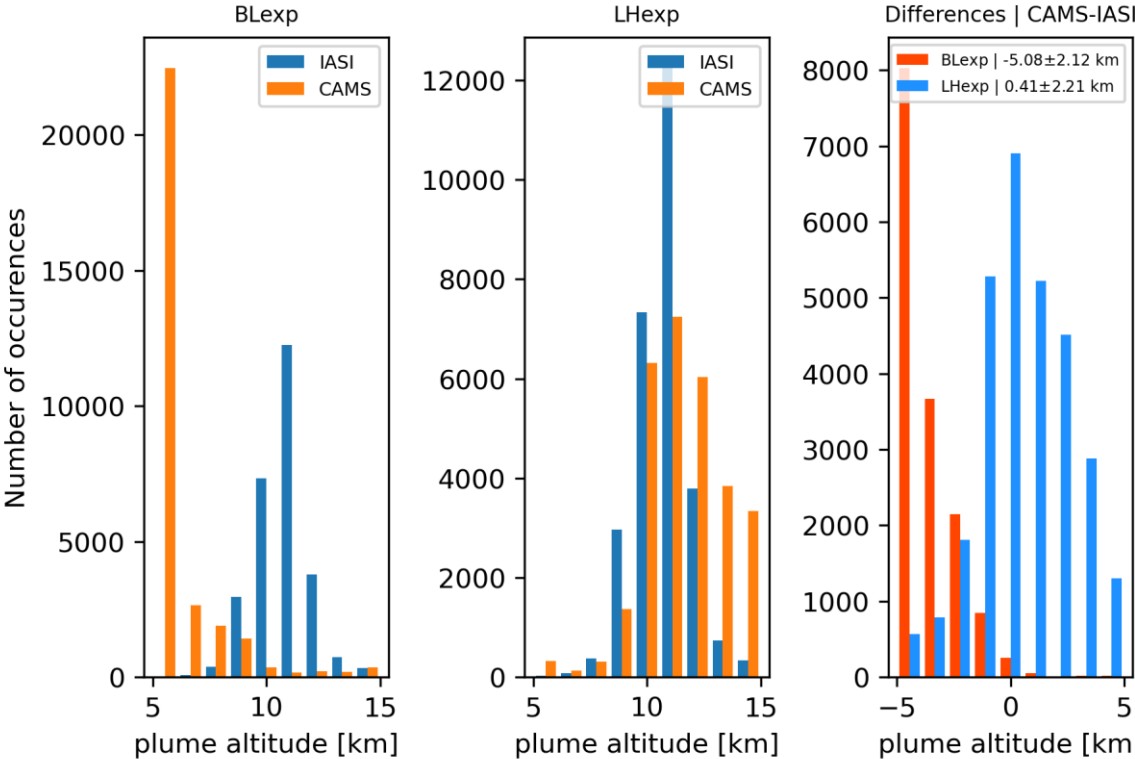

**Figure 15: Comparison of plume altitude from IASI ULB LATMOS data with the altitude of maximum SO₂ concentration from the LHexp (middle panel) and BLexp (left panel) for the period 22-29 June 2019. The right panel shows a histogram of the differences of the plume altitudes (CAMS minus IASI) for LHexp (blue) and BLexp (red).**

### 4.3.3 Quality of the 5-day TCSO₂ forecasts

Next, we assess the quality of the 5-day TCSO₂ forecasts started from the LHexp and BLexp SO₂ analyses. Figure 16 shows a timeseries of POD for a TCSO₂ threshold of 5 DU from LHexp and BLexp for NRT TROPOMI and GOME-2B for the initial SO₂ analysis and forecasts valid on the same day at different lead-times (24 to 120 hours). The figure shows that the skill decreases with increasing forecast lead-time in both experiments, but that the degradation of skill with forecast lead-time is considerably large in BLexp. For the 72-hour forecasts LHexp has POD values between 0.6 and 0.8 and even the 96-hour forecast still has values of 0.4. In contrast, BLexp only has POD values between 0.2 and 0.4 for the 72-hour forecasts

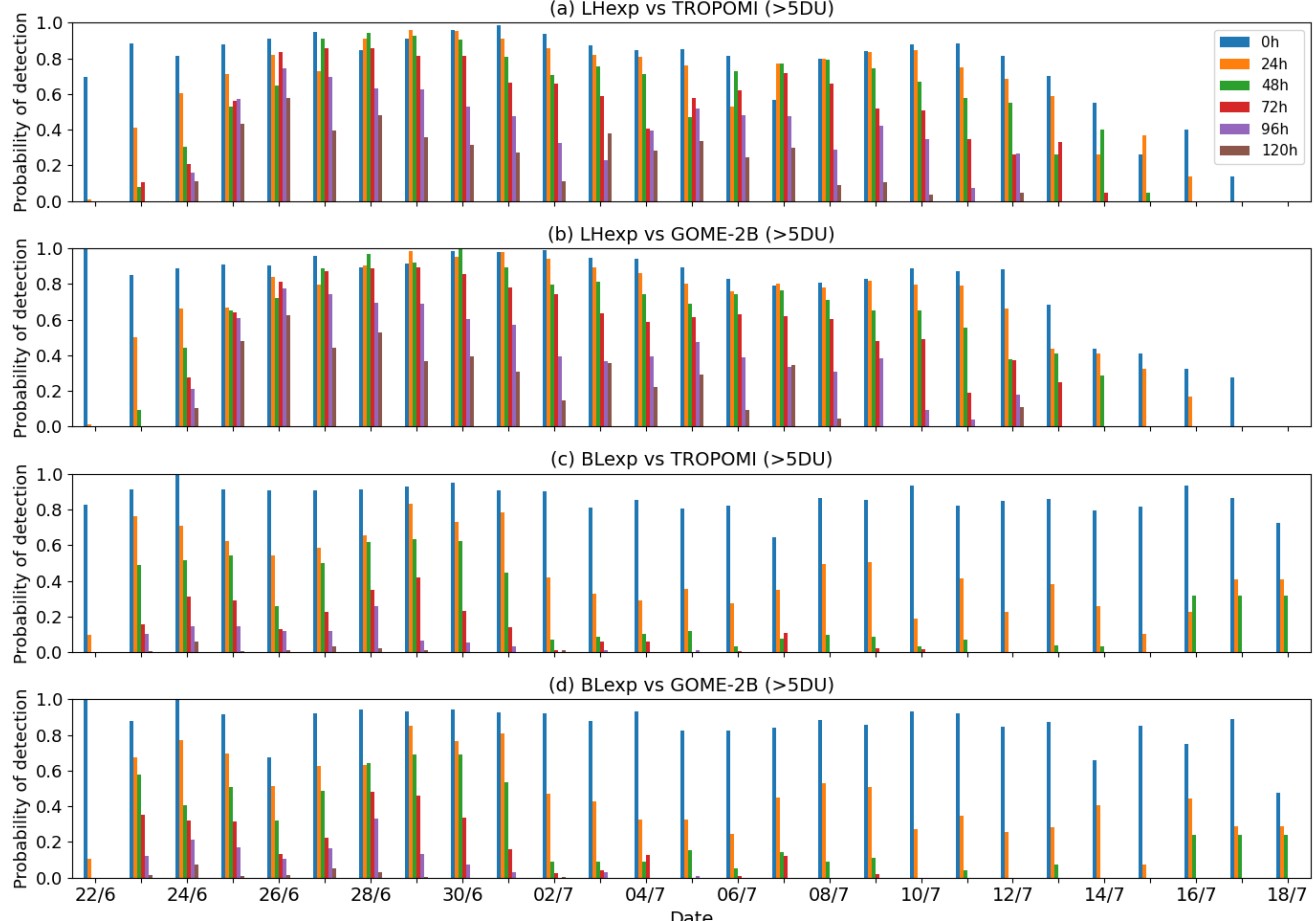

**Figure 16: Probability of detection of LHexp against (a) NRT TROPOMI and (b) GOME-2B, as well as BLexp against (c) NRT TROPOMI and (d) GOME-2B for the period 22 June to 18 July 2021 for analysis at 0z (blue), 24-hour forecast (orange), 48-hour forecast (green), 72-hour forecast (red), 96-hour forecast (purple) and 120-hour forecast (brown).**

during June while values drop considerably during July when even the short 24-hour forecasts from BLexp only have POD values between 0.2 and 0.4. In other words, in BLexp the skill of forecasting the location of the SO₂ plumes seen by GOME-

2B and the NRT TROPOMI one day in advance is similar to the skill of forecasting the SO₂ plumes 4 days in advance in LHexp. The main reason for the lower forecast quality in BLexp is the fact that the SO₂ plumes are located at the wrong altitude (see Fig. 15) so that the prevailing winds will not transport the SO₂ in the correct direction.

To further assess the forecast skill, we use the fractional skill score (FSS) which is a spatial comparison. It was originally

used to assess the quality of precipitation forecasts (Roberts and Lean, 2007) but has more recently also been used to assess the skill of dispersion models to capture volcanic plumes (de Leeuw et al, 2021; Dacre et al., 2016; Harvey and Dacre, 2016). The FSS is calculated using the ratio of the modelled and observed fractional coverage of the SO₂ plume at each location for various horizontal scales (neighbourhoods) and thresholds, and it assesses how the skill of the forecast varies depending on those parameters. To calculate it we grid the model TCSO₂ analyses and forecasts at various lead-times and the

NRT TROPOMI and GOME-2B TCSO₂ observations on a 1°x1° grid and create binary fields for the chosen thresholds (in our case for TCSO₂ > 1, 3, 5, 10, 20, 30, 50 DU). Then, for each grid point, the fraction of surrounding grid points that exceed the threshold is calculated from the model field and the observations. To establish at which horizontal scale the SO₂ analysis or forecast is useful we repeat this exercise with neighbourhoods of varying scales (i.e. 1, 3, 5°, corresponding to neighbourhoods of 1, 9 and 25 grid boxes, respectively). An FSS of 1 means perfect alignment of the features in the

observations and the model and an FSS of 0 a total mismatch. We use values of FSS greater than 0.5 to define a simulation that has some skill. This value was also used by de Leeuw et al. (2021) and Harvey and Dacre (2016). The FSS for a neighbourhood of length n is calculated following Roberts and Lean (2007) as

$$FSS_{(n)} = 1 - \frac{MSE_{(n)}}{MSE_{(n)ref}} \quad (4)$$

where MSE is the Mean Square Error and $MSE_{(n)}=0$ for a perfect forecast of neighbourhood with length n. The reference MSE for each neighbourhood length n is given by:

$$MSE_{(n)ref} = \frac{1}{N_x N_y}\left[\sum_{i-1}^{N_x}\sum_{j-1}^{N_y} O_{(n)i,j}^2 + \sum_{i-1}^{N_x}\sum_{j-1}^{N_y} M_{(n)i,j}^2\right] \quad (5)$$

Here $i=1,N_x$ with $N_x$ the number of columns in the domain and $j=1, N_y$ with $N_y$ the number of rows. $M_{(n)i,j}$ is the field of model fractions obtained from the model binary field for a square of length n and $O_{(n)i,j}$ the corresponding field of observed fractions. $MSE_{(n)ref}$ can be interpreted as the largest possible MSE that can be obtained from the model and observed fractions.

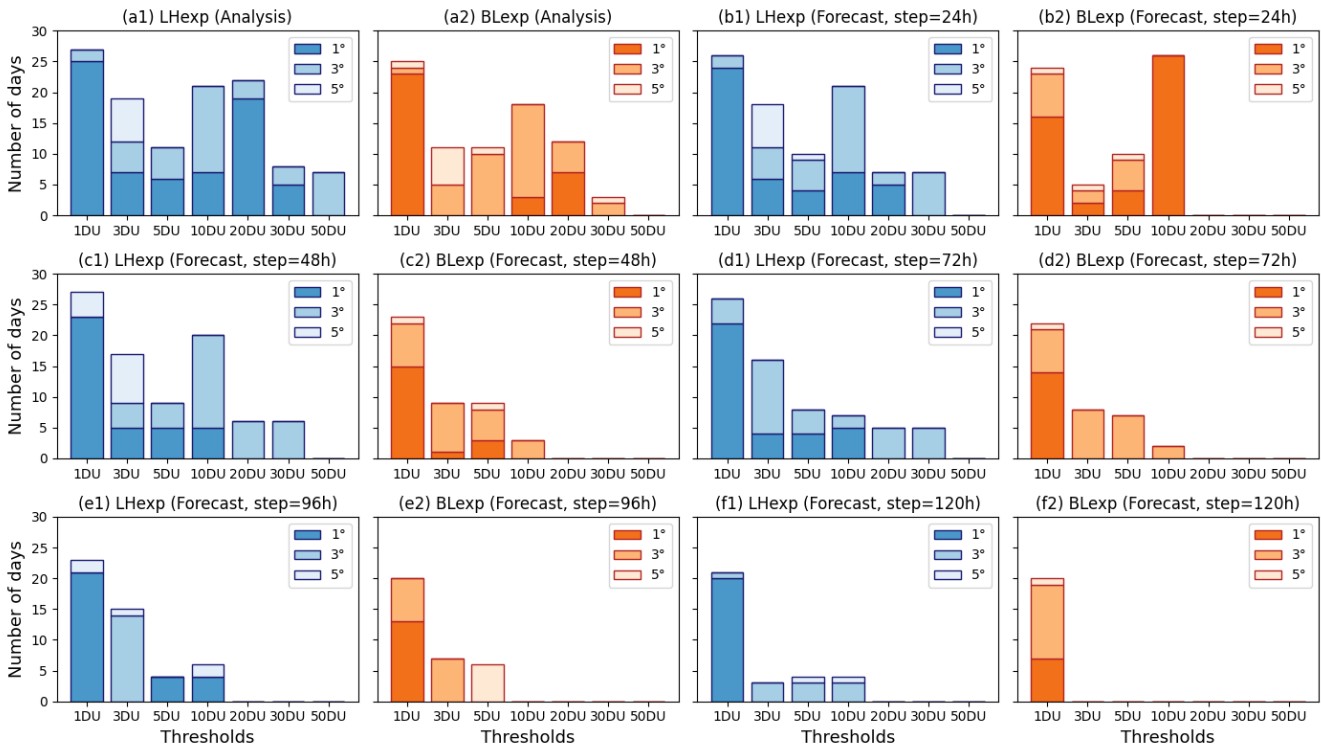

**Figure 17: Number of days since the eruption on 22 June 2019 that the LHexp (blue) and BLexp (orange) (a) analyses and forecasts at steps (b) 24, (c) 48, (d) 72, (e) 96 and (f) 120-hours have some skill (FSS>0.5) compared to NRT TROPOMI TCSO₂ data for neighbourhood sizes of 1º, 3º and 5º and thresholds of 1, 3, 5, 10, 20, 30, 50 DU.**

Figure 17 shows the number of days after the eruption that have FSS>0.5 when comparing LHexp and BLexp with NRT TROPOMI data for the various thresholds, to give some indication of a skill timescale, i.e. how long the analyses and the forecasts started from them can be considered as useful after the initial eruption. Also shown (in the lighter shadings) are the additional useful forecast days that are gained when the neighbourhood size is increased to 3° or 5°. The main findings of the figure are (1) the skill timescale is longer for the smaller thresholds, i.e. the overall shape of the plume is easier to reproduce than smaller scale filaments with higher TCSO₂ values, (2) the skill timescale drops with lead-time, (3) the skill timescale increases if the neighbourhood size is increased, with a larger increase for higher thresholds, pointing to errors in the location of structures with higher TCSO₂, which are reduced with a larger grid, and (4) that the skill timescale is greater in LHexp than in BLexp, leading to better forecasts of the plume longer in advance, as already seen in Fig. 16.

We now look at the individual panels in more detail. Figure 17a shows the skill timescales of the TCSO₂ analyses from LHexp and BLexp against NRT TROPOMI and illustrates as already seen in Section 4.3.1 that these give similarly useful TCSO₂ fields (especially for the lower thresholds), but that the number of useful days is slightly larger in LHexp and that BLexp fails to capture the highest TCSO₂ values. It is interesting to see the large number of days with FSS>0.5 for the

threshold of 20 DU in LHexp, because this is the value below which no $TCSO_2$ data are assimilated in LHexp. Figure 17b shows that the 24-hour forecasts in LHexp have similar skill to the analysis, which a skill timescale of 24 days for the 1 DU threshold and a neighbourhood size of 1°, illustrating that the 24-hour from the LHexp analysis can predict the overall location of the $SO_2$ plume very well. For higher thresholds (> 30 DU) this drops to about 5 days after the eruption, and there is no skill for a threshold of 50 DU. The skill of the 48 and 72-hour forecasts (Fig. 17c and d) are similar to that of the 24-hour one for the thresholds up to 10 DU, but at a neighbourhood size of 1° the higher values (>20 DU) have no skill anymore. As there is still skill on a 5-day timescale for these forecasts for a neighbourhood size of 3°, this suggests that it is the location of the filaments with high $TCSO_2$ values that is not correct rather than the forecast not maintaining any of the higher $TCSO_2$ values. Even the 96 and 120-hour forecasts (Fig. 17e and f) in LHexp have a skill timescale of slightly more than 20 days for the 1 DU threshold at 1°, but the skill drops markedly for the higher thresholds, and for the 120-hour forecasts skill is only found for thresholds up to 10 DU at 3° for up to 3 days after the eruption. Nevertheless, Fig. 17 shows that by assimilating $SO_2$ LH data the CAMS system can predict the overall location of the $SO_2$ plume up to 5 days in advance for about 20 days after the initial eruption. This corresponds to the time when the $SO_2$ LH product does not detect volcanic $SO_2$ anymore (see Fig. 8). Leeuw et al. (2021), using the Met Office's Numerical Atmospheric-dispersion Modelling Environment (NAME) dispersion model, found skill timescales of 12–17 days for low density (> 1 DU) parts of the Raikoke $SO_2$ cloud and shorter skill timescales of 2–4 days for the denser parts of the cloud (> 20 DU). It is interesting to see skill timescales of similar magnitude to the ones obtained in our study even though their method is different. Leeuw et al. (2021) initialized the NAME dispersion model with eruption source parameters and then followed the evolution of the $SO_2$ cloud, while we use data assimilation to update the location of the plume daily and provide daily forecasts with a maximum length of 5 days.

Figure 17 shows that the BLexp analysis has skill timescales similar to LHexp, confirming what was already seen in Figures 12 and 13. Despite placing the $SO_2$ cloud at the wrong altitude, the overall shape of the $SO_2$ plume is still captured by the $SO_2$ analysis. However, for higher thresholds the number of useful days after the eruption is smaller in BLexp and the forecast skill drops more steeply with forecast lead-time than in LHexp. There is no skill for the 24-hour forecast at 1° for thresholds greater than 20 DU, and for the 48-hour forecasts the skill timescale for a 1 DU threshold at 1° is 15 days, compared to 23 days in LHexp. The skill timescale remains around 14 days in BLexp for the 72 and 96-hour forecasts for a 1 DU threshold at 1° and then drops to 6 days at 120-hours. For the 72 to 120-hour forecasts there is no skill for the higher thresholds for a neighbourhood size of 1°, pointing to a worse misplacement of the smaller scale features of the plume with higher $TCSO_2$ values than in LHexp.

For GOME-2B (not shown) the number of useful forecast days are generally slightly lower, especially for a threshold of 1 DU which might just be an artefact because GOME-2B does not detect so many volcanic pixels with low values. For thresholds of 3-30 DU the GOME-2 results for a neighbourhood size of 1° or 3° are very similar to the TROPOMI results for all the forecast ranges, with skill timescales of about 10 days for forecast lead times up to 72 hour and around 5 days for the 96-hour forecasts. Again, the performance of BLexp is worse than of LHexp and for the 48-hour forecasts there is almost no skill in BLexp for the 1° neighbourhoods.

## 5 Conclusions

In this paper we document the procedure used to assimilate near-real time $TCSO_2$ data from the TROPOMI and GOME-2 instruments in the operational CAMS NRT data assimilation system and explore the use of TROPOMI $SO_2$ layer height data provided by the ESA-funded Sentinel-5P Innovation–$SO_2$ Layer Height Project and produced with the Full-Physics Inverse

Learning Machine algorithm (v3.1) developed by DLR. The assimilation of the FP_ILM $SO_2$ LH data was tested for the 2019 Raikoke eruption and compared with results obtained when assimilating NRT TROPOMI $TCSO_2$ data with the operational CAMS configuration.

While the operational CAMS approach of placing the $SO_2$ increment in the mid-troposphere around 550 hPa gives surprisingly good results for the $TCSO_2$ analyses and short-range forecasts in a lot of situation (including this case), the vertical distribution of $SO_2$ in the baseline analysis is clearly wrong for the Raikoke eruption which injected a copious amount of $SO_2$ into the stratosphere. By using the FP_ILM TROPOMI $SO_2$ LH data this can be much improved as comparison with the independent $SO_2$ plume heights retrieved from IASI show. While the LH experiment agrees well with

the IASI LATMOS ULB plume altitude products, with a mean bias of 0.4±2.2 km, the baseline experiment underestimates the plume altitude with a mean bias of -5.1±2.1 km. Consequently, the assimilation of the FP_ILM LH data leads to much improved $SO_2$ forecasts and should improve the usefulness of the CAMS $SO_2$ forecasts for users and also for the aviation industry.

In the baseline experiment the forecast skill drops much more for longer forecast lead-times than in the LH experiment, which is seen when comparing point skill scores such as probability of detection and critical success index and when using the fractional skill score that also assesses spatial skill. Timeseries of the Probability of Detection score show that in the baseline experiment, the skill of forecasting the location of the Raikoke $SO_2$ plume seen by GOME-2B and the NRT TROPOMI one day in advance is similar to the skill of forecasting the $SO_2$ plume 4 days in advance in LHexp. The FSS

shows that compared to NRT TROPOMI, even the 120-hour forecasts of the LH experiment have a significant skill up to 20 days after the initial eruption for the prediction of $TCSO_2$ for a 1 DU threshold and a neighbourhood size of 1°, suggesting that the overall location of the $SO_2$ plume is well reproduced. The skill is smaller for higher $TCSO_2$ thresholds (about 5 days for forecast ranges up to 96-hours on a 1° grid), illustrating that it is more difficult to accurately predict the location of areas with higher $SO_2$ columns which usually have smaller spatial scales. The skill timescale is shorter for the baseline experiment,

with values around 15 days after the initial eruption for the 1 DU threshold for forecast ranges up to 96-hours and 5 days for the 120-hours forecasts, but there is no skill for any of the higher thresholds at a neighbourhood size of 1° from 72-hour forecasts onwards. By assimilating FP_ILM $SO_2$ LH data the CAMS system can predict the overall location of the Raikoke $SO_2$ plume up to 5 days in advance for about 20 days after the initial eruption.

Our study also documents some issues of the CAMS $TCSO_2$ assimilation approach, namely the overestimation of the $SO_2$ burden and plume area by the data assimilation system, both in the operational configuration and when using the FP_ILM $SO_2$ LH data. The main reason for this overestimation is the coarse horizontal resolution used in the minimisations (currently TL95 and TL159 in the operational CAMS system) which limits the wavenumbers that can be resolved in the wavelet formulation of the $SO_2$ background errors. This in turn limits the horizontal correlation length scale that can be used for the

$SO_2$ background errors and that determines how for the increments from individual observations are spread out in the horizontal. In this paper we used TL159/TL255 as the resolutions for the minimisations, but to properly resolve small scale structures the resolutions of the minimisations would have to be even higher. Obviously, this would increase the numerical cost of running the minimisation.

Other reasons that can contribute to an overestimation of the $SO_2$ burden or plume area in the CAMS $SO_2$ analysis could be the use of anthropogenic $SO_2$ emissions in the CAMS model as the satellite data used for the comparisons are only the volcanic pixels. However, tests run without the anthropogenic emissions (not shown in this paper) did not show large differences compared to the experiments presented here, suggesting that this is not a big effect for the Raikoke eruption.

Another possibility could be the fact that the satellite might miss a plume or part of a plume, but that the whole plume is present in the model. Finally, for the FP_ILM LH product the data are limited to TCSO$_2$> 20 DU (in v3.1) and lower values that might correct an overestimation from the previous analysis cycle are not used. In future we hope to also test the assimilation of IASI SO$_2$ data with plume height information that would add extra information in the CAMS system.

One limitation in using the TROPOMI SO$_2$ LH data is that the version used in this study (v3.1) only produces reliable information for TCSO$_2$>20 DU so that most of the smaller volcanic eruptions that happen on a more regular basis than big explosive eruptions would be missed if only the FP_ILM TROPOMI SO$_2$ LH data were assimilated in the CAMS NRT system. Improvements to the TROPOMI SO$_2$ LH product are on-going so that it should be possible to lower this limit in the future.

**Code and Data availability**

This study was based on the IFS model cycle 47R1. The ECWMF IFS code is only available subject to a licence agreement with ECMWF. ECMWF member-state weather services and their approved partners will get access granted. The IFS code without modules for assimilation can be obtained for educational and academic purposes as part of the openIFS release (https://confluence.ecmwf.int/display/OIFS, last access 26/10/2021). A software licensing agreement with ECMWF is required to access the OpenIFS source distribution: despite the name it is not provided under any form of open-source software license. License agreements are free, limited to non- commercial use, forbid any real-time forecasting, and must be signed by research or  educationalorganizations. Personal licenses are not provided. OpenIFS can-not be used to produce or disseminate real-time forecast products. ECMWF has limited resources to provide support and thus may temporarily cease issuing new licenses if it is deemed too difficult to provide a satisfactory level of support. Provision of an OpenIFS software license does not include access to ECMWF computers or data archives other than public datasets.  A detailed documentation of the IFS code is available from   https://www.ecmwf.int/en/publications/ifs-documentation (last access 26/10/2021). The output from the assimilation experiments used in this study is available from https://apps.ecmwf.int/research-experiments/expver/ (last access 26/10/2021) using the following DOIs for the 6 experiments:

- hhu5: 10.21957/cygt-xf49
- hgze: 10.21957/qfam-7474
- hhbu: 10.21957/zpdt-f079
- hhtm: 10.21957/jraa-s174
- hhtn: 10.21957/ddxs-2v95
- hgz7: 10.21957/81bh-7h58

The TROPOMI V3.1 SO$_2$ LH data are available from https://doi.org/10.5281/zenodo.5602935, the operational TROPOMI SO2 data from the Copernicus Open Access Hub (https://scihub.copernicus.eu/) and the IASI SO$_2$ plume height data from https://en.aeris-data.fr/.

**Acknowledgements**

The Copernicus Atmosphere Monitoring Service is operated by the European Centre for Medium-Range Weather Forecasts on behalf of the European Commission as part of the Copernicus program (http://copernicus.eu) and CAMS data are freely available from atmosphere.copernicus.eu/data. The SO$_2$ analysis and forecast experiments used in this paper are available from https://apps.ecmwf.int/research-experiments/expver/ (see DOIs in Table 3). The FP_ILM TROPOMI SO$_2$ LH product was developed in the framework of the ESA Sentinel-5p Innovation project. MEK would like to acknowledge the Aristotle

University of Thessaloniki (AUTh) High Performance Computing Infrastructure and Resources and the help of the AUTh IT Center. MEK further acknowledges the use of the Atmospheric Toolbox®. IASI is a joint mission of EUMETSAT and the Centre National d'Etudes Spatiales (CNES, France). The authors acknowledge the AERIS data infrastructure, https://en.aeris-data.fr/, for providing access to the IASI $SO_2$ plume height data used in this study and ULB-LATMOS for the development of the retrieval algorithms. Thanks to Anabel Bowen for improving Figures 5 and 9.

## Author Contributions

AI prepared the code to assimilate the $SO_2$ LH data, ran the analysis experiments, carried out most of the validation and wrote the paper. MA helped with the construction of the background error matrices. JF provided help with the modelling framework and RR wrote the converter software to transfer the $SO_2$ LH data from their native netcdf format to the BUFR format used in the ECMWF data assimilation system. MEK and DB performed the validation against the IASI/MetOp $SO_2$ plume altitude data and provided Fig. 15. PH provided the $SO_2$ LH data and developed the FP_ILM algorithm. DE and DL also contributed to the FP_ILM developments. All co-authors provided useful feedback on the paper.

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
