# Peer review of "Evaluating the assimilation of S5P/Tropomi NRT SO2 columns and layer height data into the CAMS integrated forecasting system (CY47R1), based on a case study of the 2019 Raikoke eruption"

_Geoscientific Model Development, 2021_

## Author Comment (AC1)

**Reply to Juan A. Añel (Feosc. Mod. Dev. Exec. Editor)**

Our paper uses SO2 **data** or **products** and we often referred to them as 'retrievals' in the original version of the paper. No work with any retrieval code has been carried out for this paper, it is simply a data assimilation paper. It should be possible to reproduce the work described in this paper by just using the TROPOMI SO2 data and the model output, and no access to the TROPOMI retrieval algorithm is required.

We realize that we used the term 'retrieval' a lot when referring to the data and have made some changes to the document and replaced a lot of those 'retrievals' with 'data' or 'products' in the text. This makes it clearer that the paper is utilizing data and not developing or refining a retrieval algorithm. We have also replaced the one location where we used 'retrieval algorithm', i.e. "Improvements to the FP_ILM retrieval algorithm are on-going" with "Improvements to the TROPOMI SO2 LH product are on-going".

We have added a DOI for the TROPOMI LH data used in this study, a link to the ESA S5P website where the operational S5P SO2 data can be downloaded and additional information about the ECMWF code and availability. Note, that we had already provided DOIs for the data assimilation experiments used in this paper.

We will upload a modified version of the paper.

**Code and Data availability**

This study was based on the IFS model cycle 47R1. The ECWMF IFS code is only available subject to a licence agreement with ECMWF. ECMWF member-state weather services and their approved partners will get access granted. The IFS code without modules for assimilation can be obtained for educational and academic purposes as part of the openIFS release (https://confluence.ecmwf.int/display/OIFS, last access 26/10/2021). A software licensing agreement with ECMWF is required to access the OpenIFS source distribution: despite the name it is not provided under any form of open-source software license. License agreements are free, limited to non- commercial use, forbid any real-time forecasting, and must be signed by research or educationalorganizations. Personal licenses are not provided. OpenIFS can-not be used to produce or disseminate real-time forecast products. ECMWF has limited resources to provide support and thus may temporarily cease issuing new licenses if it is deemed too difficult to provide a satisfactory level of support. Provision of an OpenIFS software license does not include access to ECMWF computers or data archives other than public datasets.  A detailed documentation of the IFS code is available from https://www.ecmwf.int/en/publications/ifs-documentation (last access 26/10/2021). The output from the assimilation experiments used in this study is available from https://apps.ecmwf.int/research-experiments/expver/ (last access 26/10/2021) using the following DOIs for the 6 experiments:

- hhu5: 10.21957/cygt-xf49
- hgze: 10.21957/qfam-7474
- hhbu: 10.21957/zpdt-f079
- hhtm: 10.21957/jraa-s174
- hhtn: 10.21957/ddxs-2v95
- hgz7: 10.21957/81bh-7h58

The TROPOMI V3.1 $SO_2$ LH data are available from https://doi.org/10.5281/zenodo.5602935, the operational TROPOMI $SO_2$ data from the Copernicus Open Access Hub (https://scihub.copernicus.eu/) and the IASI $SO_2$ plume height data from https://en.aeris-data.fr/.

---

## Author Comment (AC2)

Replies to reviewer 1

Thanks a lot for taking the time to read the paper and giving us valuable comments. We have changed the manuscript according to the suggestions and have listed our replies and changes in blue below.
* * *
The authors present the assimilation of SO2 retrievals from Tropomi satellite observations in the global forecasting system used in CAMS for volcanic forecasting. As for other major centres, assimilating vertically-integrated information on SO2 from space-borne sensors is a challenge which needs continuous improvement, as observational product and data assimilation settings can be refined or improved year after year. This paper is of interest to the community. I suggest it is accepted after modifications are made.

**Scope and title**
The title is a bit misleading as the study presented in this manuscript is presenting assimilation experiments carried out in a different system than the near real time (NRT) CAMS system used for volcanic forecasting. Moreover, the present study mainly compares results obtained assimilating the new product proposed by the DLR including information on the SO2 plume vertical extension, with several settings, to those obtained in thecurrent operational setting with NRT Tropomi data isseminated by ESA. In addition, the study described in this manuscript only focusses on a particular eruptive event, the Raikoke 2019 eruption, which injects S02 plumes at very high altitudes. No other event is assessed in this study. Eruptive events release SO2 plume at a large range of altitudes, depending on the volcano and the given episode. The present paper does not provide any guidance for other eruptive events. I suggest to change the title so as to reflect the content of the paper more closely, such as
"Evaluation of the assimilation of the S5P-Tropomi SO2 layer height product in the CAMS global system in the case of the Raikoke 2019 eruption".

We have changed the title to:

*'Evaluation of the assimilation of S5P/Tropomi SO2 layer height data in the CAMS global system for the Raikoke 2019 volcanic eruption.'*

**Assimilation settings for the observations**
Section 3.2.1 (235)
The authors describe the baseline configuration and say "SO2 observations are currently only assimilated ... when the observed SO2 concentrations are considerably larger than the atmospheric background values". I suggest the authors clearly state that criterion, instead of vaguely referring to "considerably larger".

We already mentioned in section 2.1 :' Furthermore, only TROPOMI SO2 pixels with values greater than 5 DU are assimilated in the operational CAMS system to avoid assimilating SO2 from outgassing volcanoes which are covered by SO2 emissions in the CAMS model' . For GOME-2 we assimilate all the pixels flagged as volcanic, which is also stated in section 2.1. With the statement in line 235 we only wanted to illustrate that we can not use an NMC style method because the resulting background errors would peak at the surface where anthropogenic emissions lead to the largest SO2 values. They would not give us background error statistics which would be useful for volcanic eruptions as that information is not in the model's background field. Reading the sentence again, the part 'considerably larger…' is not really needed and we have removed it, so that the sentence now simply reads:
*'SO2 observations are currently only assimilated in the CAMS system in the event of volcanic eruptions.'*

I may have missed the description of the observation pre-processing in the paper. Can the authors state clearly how the mismatch between the observation resolution and the model resolution? Are data thinned? Is there a super-obbing step? What are the parameters of the pre-processing?

The TROPOMI data are super-obbed to the model resolution. We already mention this in Section 2.1: 'The TROPOMI SO2 data are averaged to the model resolution (TL511, about 40km) before being used in the CAMS system. '
The GOME-2 data are used at the satellite resolution which is similar to the model resolution. We have added in Section 2.3:
*'The GOME-2 data are used at the satellite resolution which is similar to the resolution of the CAMS model used in this paper.'*

As the number of observations varies between NRT and LH SO2 observations, a clear indication of the difference in the number of assimilated data should be clearly given.

We already show in Figure 8 a timeseries of the number of observations and have already this text in the paper: 'Figure 8 shows a timeseries of the number of observations that are actively assimilated in both experiments, i.e. the number of 1ºx1º grid points with active observations, and illustrates that there are more active data in BLexp where NRT TROPOMI SO2 data with values greater than 5 DU are assimilated (i.e. as done in the operational CAMS system) than in LHexp where only data with LH TCSO2 greater than 20 DU are assimilated.'

No word is said on the observation errors, which are also important players in the game. The reader would benefit from a clear description on how the observation errors are handled.

We use the observation errors given by the data providers we have added a sentence in Section *' For the TROPOMI data (and also the other SO2 products used in this paper) observation errors as given by the data providers are used.'*

NRT Tropomi SO2 observations are provided with averaging kernels. Are these averaging kernels used in the baseline configuration? Are SO2-LH observations provided with averaging kernels? If present, are the latter used in the assimilation? I suggest the authors clearly state all these "details".

The NRT Tropomi SO2 observations are indeed provided with averaging kernels. However, for the volcanic SO2 product the averaging kernels are simply 1 km box profiles that are used in the AMF calculation to represent typical volcanic SO2 profiles and do not provide any real information about the current eruption. It therefore does not make sense to use these in the CAMS assimilation system. There are 3 different averaging kernels provide for each SO2 column retrieval and the user can choose the product that best suits the situation. See TROPOMI ATBD for more information: https://sentinel.esa.int/documents/247904/2476257/Sentinel-5P-ATBD-SO2-TROPOMI.

We have added more information at the end of Section 2.1:

*The DOAS vertical column SO2 retrieval requires knowledge of a prior SO2 profile to convert the slant columns into vertical columns. Because this profile shape is generally not known at the time of the observation and it is also not know whether the observed SO2 is of volcanic origin or from pollution (or both) the TROPOMI algorithm calculates four vertical columns for different hypothetical SO2 profiles. One vertical column is provided for anthropogenic SO2 with the prior SO2 profile taken from the TM5 CTM and three for volcanic scenarios assuming the SO2 is either located in the boundary layer, in the mid-troposphere (around 7 km) or in the stratosphere (around 15 km). These volcanic prior profiles are box profiles of 1 km thickness, located at the*

*corresponding altitudes. The NRT CAMS system uses the mid-troposphere product. TROPOMI SO2 data are provided with averaging kernels based on the prior hypothetical SO2 profiles (i.e. the 1 km box profiles centred around the assumed SO2 altitude for the volcanic columns). However, as these do not provide any real information about the altitude of the volcanic plume they are not used in the CAMS system. More information about the NRT TROPOMI SO2 retrieval can be found in the TROPOMI ATBD. For the TROPOMI data (and also the other SO2 products used in this paper) observation errors as given by the data providers are used.*

**Minor comments**

line 397: data are gridded for comparison. What is the time step for this gridding: daily or hourly?

The calculation of the analysis or first-guess fields is done at the time and location of the observations in the observation operator of the model. Later, all data (obs or analysis/forecast) in a 12-hour analysis window are interpolated onto a 1x1 degree grid. We have added in section 4.1 (where we first mention the gridding):

*'All the satellite data available during a 12-hour assimilation window were gridded onto a 1⁰x1⁰ degree grid….'*

Figures showing timeseries are numerous and sometimes hardly legible (eg. 12, 13).

We have improved several of the figures, including Fig 12 and 13.

Figures showing maps are sometimes a bit small (eg. 5, 9)

We think the quality of Figures 5 and 9 is good enough for publication. The main point of the figures is to give an overview of the evolution of the SO2 plume and they are big enough for that.

Do the authors think showing evaluation for D+5 forecasts is relevant for such a study which shows the high sensitivity to the assimilation settings?

As the CAMS forecast system provides 5-day forecasts we think it is relevant to show them.

---

## Author Comment (AC3)

Thanks a lot for taking the time to read the paper and giving us valuable comments. We have changed the manuscript according to the suggestions and have listed our replies and changes in blue below.
* * *
**General comments**

This paper presents the CAMS assimilation of volcanic SO2 satellite data, and in particular improvements made to the system by the use of layer height information retrieved from satellite, which show to improve the SO2 forecasts. The paper is interesting and presents both improvements to and current challenges with the system. The topic of the paper is highly relevant as it addresses a method which can be used to fuse models and observations and targets a particular application to volcanic clouds. The paper is well written and highly suited for publication; however, I would like the below comments to first be addressed.

I miss some discussion around the applied/assumed thickness of the SO2 plume and if/how this might affect the results. See specific comment on L260.

I am concerned that the model simulations do not directly consider the vertical averaging kernel information from the SO2 retrieval. See specific comment on L262.

I miss some further details on the TROPOMI SO2 retrieval. Other TROPOMI SO2 total column retrievals are interlinked with assumptions on the SO2 plume altitude and often different products are available based on different a priori plume altitudes (e.g., the Copernicus SP5 products). L122 mention prior SO2 vertical profile shapes but this is not mentioned again or discussed any further for the DLR TROPOMI retrieval (only for IASI on L186). Please elaborate further on which prior profiles are used in the DLR TROPOMI retrievals (both NRT and LH) and if these vary, and also how/if this affects the retrieval of the layer height.

We have added this information in Section 2.1:

*The DOAS vertical column SO2 retrieval requires knowledge of a prior SO2 profile to convert the slant columns into vertical columns. Because this profile shape is generally not known at the time of the observation and it is also not know whether the observed SO2 is of volcanic origin or from pollution (or both) the TROPOMI algorithm calculates four vertical columns for different hypothetical SO2 profiles. One vertical column is provided for anthropogenic SO2 with the prior SO2 profile taken from the TM5 CTM and three for volcanic scenarios assuming the SO2 is either located in the boundary layer, in the mid-troposphere (around 7 km) or in the stratosphere (around 15 km). These volcanic prior profiles are box profiles of 1 km thickness, located at the corresponding altitudes. The NRT CAMS system uses the mid-troposphere product. TROPOMI SO2 data are provided with averaging kernels based on the prior hypothetical SO2 profiles (i.e. the 1 km box profiles centred around the assumed SO2 altitude for the volcanic columns). However, as these do not provide any real information about the altitude of the volcanic plume they are not used in the CAMS system. More information about the NRT TROPOMI SO2 retrieval can be found in the TROPOMI ATBD. For the TROPOMI data (and also the other SO2 products used in this paper) observation errors as given by the data providers are used.*

We have addressed the other comments in the list of specific comments below.

Can you provide some indications as to how much more expensive (in terms of run time) the model runs are with the higher spectral resolutions used? For example, it would be very interesting to know the difference in run time for each of the experiments in Table 3.

The experiments shown in Table 3 all use the same spectral resolution (model at T511, minimisations at T159/T255) so it does not make sense to add any run time information. Compared to the operational configuration which uses T95/T159 spectral resolutions in the minimisation the numerical cost for one analysis cycle is increased by about 20-30%, with the largest increase from the second

minimisation which is about 50% more expensive when going from T159 to T255. We have added this text:

*' The numerical cost of one analysis cycle increases by about 20-30% when the spectral resolution of the minimisation is increased in this way, with the largest increase coming from the second minimisation which is about 50% numerically more expensive.'*

Specific comments
L30 - the last sentence of the abstract: It would be good to include here something about the increase in skill time scales by including the LH information. I would also include a couple more key results here; that including LH information leads to higher modelled TCSO2 values in better agreement with the satellite observations, but that plume area and burden are overestimated also when including LH data and that the reason for this overestimation is explored.

We have added this to the abstract:
*Including the layer height information leads to higher modelled TCSO2 values in better agreement with the satellite observations. However, the plume area and SO2 burden are generally overestimated in the CAMS analysis also when LH data are used. The main reason for this overestimation is the coarse horizontal resolution used in the minimisations. By assimilating the SO2 layer height data the CAMS system can predict the overall location of the Raikoke SO2 plume up to 5 days in advance for about 20 days after the initial eruption which is better than what is obtained with the operational CAMS configuration (without prior knowledge of the plume height) where the forecast skill drops much more for longer forecast lead-times.*

L40:" SO2 in the aircraft cabin is the biggest issue leading to respiratory problems for passengers and crew". Respiratory problems related to SO2 depend on the SO2 concentrations/dose and air quality standards for SO2 exist. Potential problems also depend on people's underlying health problems like asthma. It doesn't therefore always lead to respiratory problems as this sentence seem to indicate.

We have changed the sentence to: '*SO2 in the aircraft cabin is the biggest issue and can lead to respiratory problems….'*

L80: You use the different terms 'injection height' / 'plume height' / 'layer height' but not consistently and the difference between them (if any) is not explained. Personally, I'd use injection height as above the volcano and plume/layer height for the cloud altitude away from the vent, but it might be best to keep to as few terms as possible throughout the paper.

In L80 it makes sense to keep injection height, because this is what is determined by Flemming and Inness (2013). We have modified the rest of the paper to not use injection height and only use either layer height or plume height.

L135-147 (section 2.2): I miss some details on how the retrieval of the LH is done and what it relies on besides the exact wavelength ranges used. It which cases does it work well and which not (see related comment on L416). Why does it not work well below 20 DU? Also, what does this LH mean physically? You later use the height of the modelled maximum concentration as the model equivalent, would be good to comment on this here to justify that that is appropriate.

For SO2 columns below 20 DU the error of the retrieval of the layer height gets larger, making the data less accurate and less useful. The retrieval algorithm is documented in detail in other papers that we refer to. We have added a sentence in Section 2.2 and also a reference to a new validation paper by Koukouli et al. which has just been submitted to ACP:

*For low SO2 columns, high-altitude layer heights cannot be retrieved and the retrieval is biased towards low layer heights (Hedelt et al., 2018). Therefore, the use of the data in the CAMS system is restricted to values > 20 DU. More details about the retrieval algorithm can be found in Hedelt et al. (2018) and Koukouli et al. (2021). Koukouli et al. (2021) compared the S5P LH data with IASI observations for the 2019 Raikoke, the 2020 Nishinoshima and the 2021 La Soufrière-St Vincent eruptive periods and found good agreement with a mean difference of ~0.5±3km, while for the 2020 Taal eruption, a larger difference of between 3 and 4±3km was found.*

L207/section 3.2: The reader needs to know quite a bit about 4DVar assimilation systems to follow this section. It would be good to expand a little in particular on those aspects which you later explore in more detail: background error covariance matrix and the minimisations. Also, observations errors are not mentioned at all, how are errors in the observations taken into account?

The treatment of the background error formulation is already described in detail in section 3.2.1. We have added more information about 4D-var in Section 3.2:
*In the CAMS 4D-Var a cost function that measures the differences between the model's background fields and the observations is minimized to obtain the best possible forecast through the length of the assimilation window by adjusting the initial conditions.*

We have added a sentence in Section 2.1 to state that we use the observation errors provided by the data providers.

L260: "calculate the SO2 column not between the surface and the top of the atmosphere, but between the pressure values that correspond to the bottom and the top of the retrieved volcanic SO2 layer. The depth of this layer is currently set in the FP_ILM retrieval as 2 km, which corresponds to the uncertainty of the retrieved layer height." I am a little confused about this. Does it mean you use a fixed plume thickness of 2 km to calculate the modelled total columns, i.e., that you only calculate the SO2 column between the bottom of the plume (retrieved LH – 2 km) and the retrieved LH? What if there is a much thicker plume say several km thick, then the calculation of the SO2 column loading will miss a large fraction of the SO2 in the vertical by only summing only over the LH-2km depth.

You are right, the observations assume a depth of the layer of 2 km and we use this to calculate the model equivalent in the observation operator. If the SO2 layer was deeper this would not be accounted for in our method, but as that information is also not available from the observations we use it is not possible to include it based on assimilation of the SO2 LH product alone. Additional data (lidar?) which would give vertically resolved information would be needed. Some vertical variation in the SO2 loading will be achieved if parts of the plume have different altitudes as this information will be available from the observations. As you can see in Figure 3 there is quite a spread in retrieved LH for the eruption and that information will be brought into the SO2 analysis. Apart from that, we will depend on vertical transport to modify the vertical SO2 distribution.

We have added this sentence to the paper to document the limitation: '*One limitation of this method is that the SO2 LH product gives the plume altitude with an accuracy of 2 km, but does not give a value for the lower vertical boundary of the SO2 plume, and for a thick plume part of the SO2 loading could be missed in the calculation of the model equivalent. However, as the model's background SO2 concentrations in the free troposphere are low this should not be a big issue in the column calculation. Also, some vertical variation of the SO2 loading will be achieved if parts of the plume have different altitudes, and Figure 3 shows that this is indeed the case for the Raikoke eruption.*'

L262: "This approach mimics the procedure of using averaging kernels with box profiles given for the SO2 layer.". I don't understand how this mimic the use of averaging kernels

because if applying an averaging kernel sensitivity, the model data would be multiplied with a different sensitivity/ averaging kernel (AK) value at different vertical levels. Please elaborate. Ideally the satellites vertical AK profiles should be applied to the model data prior to any comparisons to the satellite data – this AK profile can vary from one satellite pixel to the next.

It mimics the use of the averaging kernels that are supplied with the volcanic SO2 data, which are supplied with the TROPOMI data and are box profiles (see our reply to reviewer 1 above) that represent typical volcanic SO2 profiles and are used in the AMF calculation to calculate the vertical columns.

We have added information about the AK in section 2.1 and changed the sentence to:
*This approach mimics the procedure of using TROPOMI SO2 averaging kernels which are box profiles, but for the retrieved layer and not an assumed hypothetical volcanic SO2 profile (see TROPOMI SO2 ATBD, http://www.tropomi.eu/documents/).*

L278: "The 'dip' in the TROPOMI SO2 burden after the initial peak is an artefact that results from missing observations in the TROPOMI NRT data." This 'dip' is not seen in the equivalent time series shown in the de Leeuw paper (their Fig 11) which also show TROPOMI data (different retrieval method). What is the cause of these 'missing observations?

In the NRT data we are using there is a data gap in the area of highest SO2 values (also visible in Figure 9c2) on 25 June. We do not know why. Possible cloud/ ash contamination or data being flagged because of 'unrealistically' high SO2 columns? De Leeuw et al. (2021) do not mention that they used NRT TROPOMI SO2 data, so we assume they used an offline product for which that problem might have been corrected. We have added this information in the paper:
*The 'dip' in the TROPOMI SO2 burden after the initial peak is an artefact that results from missing observations in the TROPOMI NRT data on 25 June 2019 in the area of highest SO2 values (also visible in Figure 9c2 below).*

L300 / Table 3: From the order of the experiments given in the table I expected first the difference between the BLexp and LHexp to be discussed, however the LH50/100/250 cases are first discussed. Perhaps guide the reader at the start of the section to say which experiments are compared first and why. Similarly, would be good there to guide the reader to say that the BLexp and LHexp will be further explored later to assess the skill timescales to see if using a more realistic height rather than the default 5 km will improve the forecasts – a key point and question for the paper.

We think this is addressed in the paper when introducing Table 3 because we already have a paragraph describing the experiments: *'...listed in Table 3. The baseline experiment (BLexp) which assimilated NRT TROPOMI TCSO2 data with the operational CAMS configuration and the layer height experiment (LHexp) which uses the FP_ILM S5P LH data with a horizontal background error correlation length of 100 km and background error standard deviation values of 0.7e-7 kg/kg are the main experiments used in this paper (Section 4.3 below) to assess if the assimilation of the SO2 LH data* using a more realistic height rather than the default 5 km *improves the CAMS SO2 analyses and forecasts. The other LH experiments assess the impact of using different horizontal SO2 background error correlation length scales and various SO2 background error standard deviation values.'*
We have added the green part to make it even clearer.

We can change the order of the entries in the table if the editor deems this necessary.

L415: "TROPOMI NRT lower detection limit": is this a true detection limit from the sensor/retrieval or do you mean rather than you applied a lower DU threshold (5 DU) for the NRT TROPOMI data compared to the SP ILM SO2LH retrieval data (20 DU)? Not clear to me if this is a direct 'detection limit' or more a 'chosen threshold' based on various

limitations (not necessarily a detection limit). For the 5 DU threshold you mention this is applied to avoid assimilating SO2 from outgassing volcanoes which are covered by SO2 emissions in the CAMS model. Also see related question below.

This is a real detection limit. The plots of the NRT TROPOMI data show all available volcanic NRT SO2 data even though only values > 5DU are assimilated.

L416: "FP_ILM SO2LH retrieval (v3.1) does not provide reliable information for TCSO2 < 20 DU and therefore only picks up those parts of the plume that are associated with the highest SO2 load" The work 'information' is ambiguous. Does it mean that both the retrieved column load values and the layer height values are not reliable under 20 DU, or is it only the retrieved layer height data which is not reliable under 20 DU? Maybe to add in section 2.2.

It means the layer height retrieval has too large an error to be useful. We have added more information in Section 2.2. See reply to L135-147 above.

L425/ Figure 9: Would be useful if the figure caption could explain why the NRT TROPOMI data differ to the SO2LH TROPOMI data (i.e., DU levels used/displayed).
We already mention this in the text, but have now added in the caption of Fig 9:
*'In panels (c)-(e) all available observations are shown, illustrating that the SO2 LH product only picks up those parts of the plume that are associated with the highest SO2 load.'*

L430: It is not directly explained why the SO2 burden is so much larger (2-3 Tg) for the LHexp compared with BLexp. Is it because of higher TCSO2 values as well as overestimating the plume area? 2-3 Tg is quite a lot higher than the total burden values from the satellite data.

The overestimation of the plume area in LHexp is actually less than in BLexp for >5 DU. The larger overestimation of the burden in LHexp is likely the result of differences in the background error standard deviation values and the fact that lower SO2 columns that could correct an overestimation in parts of the plume are not assimilated. We have added this sentence to the paper:
*The larger overestimation of the SO2 burden in LHexp is the result of differences in the background error standard deviation values used in the experiments and of the fact that lower SO2 columns, which could correct an overestimation in parts of the plume, are not assimilated.*

L575-L590: It would be good to compare these skill time scales to what was found by de Leeuw et al for the NAME model (skill for 12-17 days for the low-density (<1 DU) parts of the SO2 cloud and 2-4 days for the denser parts (>20 DU) of the SO2 cloud).

We have added a sentence to this section:
*Leeuw et al. (2021), using the Met Office's Numerical Atmospheric-dispersion Modelling Environment (NAME) dispersion model, found skill timescales of 12–17 days for low density (> 1 DU) parts of the Raikoke SO2 cloud and shorter skill timescales of 2–4 days for the denser parts of the cloud (>20 DU). It is interesting to see skill timescales of similar magnitude to the ones obtained in our study even though the method is different. Leeuw et al. (2021) initialized the NAME dispersion model with eruption source parameters and then followed the evolution of the SO2 cloud, while we use data assimilation to update the location of the plume daily and provide daily SO2 forecasts with a maximum length of 5 days.*

**Technical comments**
Figure text and labels need to be increased as on a print-out version some figures (especially figures 4, 12, 13, 16,17) are very hard or near to impossible to read.

We have improved the figures.

Figure 3: Suggest changing the colour scale as there are very few values >100 DU so hard to distinguish the dots. Is this showing values only >20 DU as you mention the retrieval is accurate only for larger DU values.

We have changed the colour scale to only show values up to 250 DU.

The de Leeuw reference should be updated to the final revised version for 2021.

Done.

The reference Prata et al. 2019 is used in the main text but is not in the reference list.

Added.

Figure 18 could be removed as there are many figures and the difference to Fig 17 is not very big so describing by words in text should be sufficient.

We have removed Figure 18 but kept the text referring to the GOME-2 result.

---

## Author Response (AR2)

*Reply to editor*

Thanks a lot for the comments to the revised version of our paper. We have changed the manuscript according your suggestions and have listed our replies and changes in blue below. We hope that it is now acceptable for publication.
* * *
Editor's comments to the author:

I have read through the reviewer comments, and the revisions the authors have made, and whereas these certainly address each of the points raised, and make appropriate edits to the text and Figures accordingly, in some cases the text added is not specific enough, and requires some improvements.

Also, whilst the revised title the reviewer suggested is generally OK, it omits one of the main values of the study, in documenting and evaluating the updated baseline capability for assimilating total-column-SO2, now taking in observations from Tropomi. Although the operational volcanic SO2 plume forecasting system was described quite comprehensively by Flemming and Inness (2013), the initial volcanic forecasting system there was 8 years ago now, and assimilated OMI, GOME-2 and SCIAMCHY SO2. The updated system described here assimilates Tropomi SO2, and the manuscript sets out some detailed specifics of the method for combining into the system.

For this reason, the first of my specific comments in the Decision-Stage Topical Editor review is to request a minor revision to the title, adding "column SO2 and NRT" into the title, to highlight that both the column-SO2 and the layer-height product aspects are "new" since the original Flemming and Inness (2013) manuscript.

The only other of my major comments (see comment 4) is that section 3.1 of the manuscript "CAMS model" needs to be extended, to communicate also the important specifics of the atmosphere model being used within the particular version of the Integrated Forecasting System being used in this analysis. Reviewer 1 points out that the experiments are carried out in a different system than the near real time (NRT) CAMS system used operationally for volcanic forecasting.

I must admit that, from reading the manuscript, this is not apparent, and I was not aware of it when I carried out my initial Topical Editor review. This is an example where the specifics of the broader modelling system used in a study need to be set out clearly, and this being something that partly motivates why GMD required the specific version numbers to be communicated at the outset, in the title of the manuscript/article. The 3rd & 4[th] of my comments then requests an extra paragraph be added to section 3.1 to set out the specifics of the IFS model used in the experiments -- and also to consider changing the "The CAMS model" in section 3 title and the subsection 3.1 title to something more specific such as "The ECMWF Integrated Forecasting System" or similar.

The remaining revisions are very minor however, and I expect the authors will be able to make these changes easily -- to then enable the manuscript to progress to publication in the GMD, within the joint ACP/GMD/AMT special issue on the Raikoke eruption.

Note that line numbers below relate to the revised manuscript not the Author-Tracked-Changes manuscript.

Specific comments:
* * *
1) Title -- As I mentioned above, in the revised manuscript the authors have completely replaced the original title of the article, following exactly the suggestion of Reviewer 1. The revised title mentions only the assimilation of the SO2 layer height in the system, and although I agree that is the main focus of the article, since this is (to my knowledge) the first time the ECMWF volcanic SO2 plume forecasting system has been described with the assimilation of Tropomi SO2 data, it's important to mention this aspect of the paper also.

Suggest therefore to insert "column SO2 and NRT" after "S5P/Tropomi", also deleting "data" from the title. Also suggest to insert "forecasting" between "global" and "system".

Suggest also a slight re-wording to improve the grammar and readibility -- change "Evaluation of" to "Evaluating the", and "in the " to "into the", changing also "for the" to ", based on case study of the", finally re-ordering "the Raikoke 2019 eruption" to "the 2019 Raikoke eruption" (putting the year before the name of the volcano is the correct way to refer to a specific eruption).

I mean the title then to be "Evaluating the assimilation of S5P/Tropomi column SO2 and NRT layer height into the CAMS global system, based on case study of the 2019 Raikoke eruption"

2) Title -- this is a follow-on comment, in relation to the GMD policy to request authors state explicitly in the title the specific version number for modeling systems used in the analysis. With the revised title, "the CAMS global system" needs to have a corresponding version number applied. As in my general comments above, I think rather than the "CAMS global system", perhaps better to refer to the "Integrated Forecasting System" -- and then I think providing the IFS cycle number would then satisfy the GMD policy on version numbers.

*We have changed the title to:*

*Evaluating the assimilation of S5P/Tropomi NRT SO2 columns and layer height data into the CAMS integrated forecasting system (CY47R1), based on a case study of the 2019 Raikoke eruption*

3) Titles to Section 3 and section 3.1 -- related to comment 2, and explained in my general comments above, please change "The CAMS system" in the title to section 3 to be more descriptive -- certainly the word "global" is needed -- but as I suggested, maybe the "Integrated Forecasting System" is better in this case? And then the title of sub-section 3.1 also to follow that revision of "CAMS system". In my initial Topical Editor review I suggested the change to "volcanic forecasting system" -- and I note in Flemming and Inness (2013), that initial system was referred to as "volcanic SO2 plume forecasting system" -- and that would be my preference, for the title of this section 3 and subsection 3.1 at least. Whether the authors feel a change from "CAMS global system" to "CAMS volcanic SO2 plume forecasting system" is also appropriate I leave to them to decide. Given the GMD policy re: version numbers, perhaps better to have that as Integrated Forecasting System, but perhaps more suitable in this section 3 to re-iterate the "volcanic SO2 plume forecasting" capability -- with then the specifics of the IFS cycle used etc. then explained in the extra para added to section 3.1 (see comment 4 below).

*We have changed the section titles to:*

*3 CAMS global integrated forecasting and data assimilation system*

*3.1 CAMS volcanic SO2 plume forecasting system*

4) Add extra paragraph to section 3.1 to briefly describe the specifics of the atmosphere model used.

As in my general comments, please add an extra paragraph here to communicate the specifics of the modelling system used. This can be taken from a description of the specific releas cycle of the Integrated Forecasting System that is used in these model experiments. I note that the 2013 Flemming and Innes JGR paper included (section 3.1) quite a detailed description of the Integrated Forecasting System, and resolution of the model experiments etc. Please provide similar (but perhaps briefer, to 1 paragraph) specifics of the IFS model and resolution used in these experiments.

We have added in section 3.1:

The model version used in this paper is based on the IFS model cycle 47R1 (CY47R1, www.ecmwf.int/en/forecasts/documentation-and-support/changes-ecmwf-model), which was the operational CAMS cycle from 6 October 2020 to 18 May 2021. In CY47R1, the CAMS system uses the CAMS-GLOBANTv4.2 anthropogenic emissions (Granier et al., 2019) which include anthropogenic SO2, as well as a climatology of SO2 outgassing volcanic emissions based on satellite data (Carn et al., 2016). Further updates relative to the previous version (CY46R1) are

- change to Global Fire Assimilation System (GFAS) v1.4 biomass-burning emissions

- the exclusion of agricultural waste burning from CAMS_GLOB_ANT to avoid double-counting with GFAS

- improved diurnal cycle and vertical profile for anthropogenic emissions

- introduction of Hybrid Linear Ozone (HLO) scheme, a Cariolle-type linear parameterisation of stratospheric ozone chemistry using the multi-year mean of the CAMS reanalysis as mean state

- updated dust source function, which reduces the overestimation of dust in the Sahara, Middle East and other regions, and restores missing dust over Australia

- new sea-salt emission scheme based on Albert et al. (2016), which provides better agreement with measured sea-salt size distribution

- revised coefficients in UV processor, based on ATLAS3 spectrum.

5) Section 2.1 (lines 132-133) -- The 3rd of reviewer 1's comments asks the authors whether there is a "super-obbing" step when the total-column SO2 data is assimilated, and also whether the data are "thinned", furthermore about specifics of any pre-processing. The authors replied pointing to the text they have already, but for Trompomi that only says "The TROPOMI SO2 data are averaged to the model resolution (TL511, about 40km) before being used in the CAMS system".

I am not at all clear what is meant by "super-obbing" but the reviewer's question suggests they're requesting more information than simply that the data are averaged. If there is any further "thinning" of the data, or pre-processing in this process, some specific mention should be made of this, or else to state "without any further pre-processing or data thinning" if that is the case.

We have changed the sentence to:

The TROPOMI SO$_2$ data are super-obbed, i.e. in a pre-processing step area means are created by averaging all data (observation values as well as errors) in a model grid box to the model resolution (TL511, about 40km). These super-observations are then used in the CAMS system without further thinning.

6) Section 2.1 (lines 135-136) -- this sentence was added in response to reviewer 1's 5th comment -- I just take issue with the word "knowledge" here. It's true that the system needs to use a prior SO2 vertical profile -- but perhaps it only needs to be approximate to broadly represent the particular class of eruption. The word "knowledge" suggests a more precise profile for the eruption is needed, but that may well not be the case -- it might not be sensitive to the specifics of the profile used -- it may adjust subsequently to be reasonably accurate even with only an approximate vertical profile.

Suggest to replace "knowledge of a prior SO2 profile" instead to "an assumption for the SO2 vertical profile".

Done.

7) Section 2.1 (lines 136-138) -- Starting a sentence "Because" is poor grammar. Replace with "Since".

Done.

8) Section 3.2.1 (lines 143-144) -- this sentence (beginning "However..") was also added re: reviewer 1's 5th comment. The authors have written "these do not provide any real information". Please re-word this to be specific to what is meant.

We have reordered the sentence and it now reads:

*However, as these do not provide information about the real altitude of a specific volcanic plume they are not used in the CAMS system*

9) Section 3.2.1 (lines 144-145) -- I am not familiar with the acronym "ATBD" -- please provide the full name here. If the acronym is used later in the manuscript, also add "(ATBD)" -- but otherwise the acronym is not needed.

This is already defined in Section 2.1 "...and further information can be found in Algorithm Theoretical Basis Document (ATBD)..."

10) Section 2.1 (lines 145-146) --- this sentence was added in response to reviewer 1's 4th comment -- I think there needs to be a few extra words added to be more specific than "observation errors as given by the data providers are used". I mean to clarify exactly how they are used and where -- I guess you mean within the 4Dvar data assimilation, right? If so please add a few words after "used" to clarify the specifics of where in the assimilation process those obs-errors are used.

We added that sentence to make clear that we don't modify the observation errors in any way, but use what is given by the data providers. We have now changed the sentence to:

*For the TROPOMI data (and also the other SO2 products used in this paper) observation errors as given by the data providers are used within the CAMS data assimilation system.*

11) Section 4 (line 306) -- Change "All the satellite data available" to "All the TCSO2 satellite data available".

Done.

12) Section 4 (lines 371-373) -- The authors have added this sentence in response to the 2nd of reviewer 2's comments, but the wording here needs to be improved. Please change "The numerical cost" to the "The CPU cost" or "The computational cost" and ", with the largest increase coming from" changed to ", with ~50% of the increased cost coming from", then enabling also to delete the text "which is about 50% numerically more expensive" from the end of the sentence. I'm assuming this change is consistent with what the authors meant -- but please amend slightly if you meant something different. Main thing is that "numerical cost" needs to be "computational cost" or "CPU cost". Also suggest to insert "stage of the" after "second" and before "minimisation" -- if that's what was meant?

*We have changed the sentence to:*

*The computational cost of one analysis cycle increases by about 20-30% when the spectral resolution of the minimisation is increased in this way, with the largest increase coming from the second minimisation which is about 50% computationally more expensive than at lower resolution.*

13) Abstract (lines 29-35) -- the grammar in this new text needs improving -- "which is better than what is obtained" is colloquial English, and not appropriate for a manuscript article. Suggest to delete "what is" from that text, and also add a comma after "the initial eruption". Similarly "forecast skill drops" is colloquial -- change "drops" to "reduces".

*We have changed the sentence to:*

*By assimilating the SO2 layer height data the CAMS system can predict the overall location of the Raikoke SO2 plume up to 5 days in advance for about 20 days after the initial eruption, which is better than with the operational CAMS configuration (without prior knowledge of the plume height) where the forecast skill reduces much more for longer forecast lead-times.*

14) Introduction (lines 44-45) -- This sentence beginning "In the short term" needs revising -- the sentence is explaining a potential health hazard of SO2 in the aircraft cabin. I think the term "short term" is referring to the type of exposure, short-term exposure and long-term exposure, and suggest to replace "In the short term, SO2 in the aircraft cabin" instead with "If sufficient SO2 is diffused into the aircraft cabin", and replace "is the biggest issue and can lead to" instead to "this could potentially lead to".

*We have changed the sentence to:*

*If sufficient SO2 is diffused into the aircraft cabin this could potentially lead to respiratory problems for passengers and crew.*

15) Introduction (lines 59-60) -- This short sentence states that assimilating volcanic SO2 in NRT "is difficult". I'm not sure it adds too much though, and needs either to be deleted or to make some specific about which aspect is difficult. I'd recommend the former.

*We have removed the sentence.*

16) Section 3.2 (lines 233-236) -- This sentence was added in reply to one of Reviewer 2's comments. Change "cost function that measures the differences" to "cost function for the total difference". Also, suggest to delete the word "background" here -- you have "model's background fields" -- but it's the actual model fields that are considered here -- I get that there are specific terms in the assimilation referring to "background fields" etc. -- but when combined with "model" that is

confusing to the more general reader because they might be thinking about the non-volcanic component of the model SO2. Suggest simply to delete the word "background" here and change "model's" to "model". See for example on line 294 (section 3.2.2) the text states "the model's background SO2 concentrations" and there it means the non-volcanic part.

Done.

Also, you've written "by adjusting the initial conditions" -- but is that actually what's done in the 4DVAR DA system -- isn't it more to identify the solution that gives that minimum in the cost function? Suggest to delete "by adjusting the initial conditions". Or if I am misunderstanding, please reply to explain this to me.

You misunderstand this. In the current setup of the CAMS and ECMWF 4D-Var the cost function is minimised by adjusting the initial conditions, i.e. we have two representations of the atmospheric state over the 12-hour assimilation windows (the observations and the background (=short range model forecast)) and look for the initial state of the atmosphere at the beginning of the assimilation window that provides the best possible forecast, i.e. for which the cost function is a minimum. CAMS is also developing a prototype 4D-var system where emissions are adjusted in addition to the initial conditions. There we have a larger control vector that includes the initial state and emission scaling factors and that cost function is then minimised with respect to the initial conditions and the emission scaling factors.

17) Section 3.2 (lines 294-295) -- you've added this sentence in response to another of Reviewer 2's comments. But you've written "some vertical variation of the SO2 loading will be achieved if parts of the plume have different altitudes" . Re-word this to more carefully explain what you're referring to here. I think you mean that volcanic plumes in the troposphere tend to become deformed into multiple layers at different altitudes. Certainly the text "will be achieved" needs to be changed to something different.

Suggest maybe "We note however that some parts of the SO2 plume are likely to be deformed into multiple layers at different altitudes..." or similar text to explain better what you're communicating here.

We mean that for larger plumes the observations will give different plume altitudes if there are variations within the plume. We have changed the sentence to:

Also, some vertical variation of the SO2 loading will result from assimilating observations with varying plume altitudes for larger volcanic plumes that are not uniform in height everywhere,…

---

## Author Response (AR3)

***Reply to editor's comments from 26/12/2021:***

We have added several sentences to section 3.1 to make it clearer that we are not using a simplified SO2 tracer in this study but the SO2 tracer from the comprehensive CB05 chemistry scheme:

*In the original version of the volcanic SO2 plume forecasting system described by Flemming and Inness (2013), there was a dedicated "volcanic SO2 tracer", with oxidation based on a simple fixed timescale approach. By contrast, in the progression of the volcanic SO2 system described here, the volcanic SO2 emissions, and data assimilation of SO2, is applied to the SO2 tracer within the CB05 chemistry scheme (Flemming et al., 2015), with oxidation to sulphate aerosol occurring, based on the kinetics specified in the chemistry scheme. The main SO2 loss in the coupled chemistry-aerosol system is the conversion to sulfuric acid. There are two pathways for this (i) in the gas phase via the hydroxyl radical (OH) and (ii) in clouds (aqueous phase). Pathway (i) via OH is the main pathway in the stratosphere. Heterogenous conversion on ash particles, is not directly modelled in the IFS. Other loss processes are wet deposition and surface dry deposition. As described in Flemming et al. (2015) the IFS uses a semi-Lagrangian advection scheme. Since the semi-Lagrangian advection does not formally conserve mass, a global mass fixer is applied to the chemical tracers, including to the SO2 tracer, and a proportional mass fixer as described in Diamantakis and Flemming (2014) was used for the runs presented in this paper. More details about the CB05 chemistry scheme can be found in Flemming et al. (2015, 2017), Remy et al. (2018) and Huijnen et al. (2019).*

---

## Author Response (AR4)

***Reply to editor's comments from 4/1/2022:***

We have modified Section 3.1 according to the suggestions. The relevant paragraph now reads:

*In the original version of the volcanic SO2 plume forecasting system described by Flemming and Inness (2013), there was a dedicated "volcanic SO2 tracer", with oxidation based on a simple fixed timescale approach. By contrast, in the progression of the volcanic SO2 system described here, the volcanic SO2 emissions, and data assimilation of SO2, is applied to the SO2 tracer within the CB05 chemistry scheme (Flemming et al., 2015), with oxidation to sulphate aerosol occurring, based on the kinetics specified in the chemistry scheme. There are two pathways for this (i) in the gas phase via the hydroxyl radical (OH) and (ii) within cloud droplets (aqueous phase), with only pathway (i) occurring in the stratosphere (in the model). In the troposphere, the model includes also the SO2 loss processes of wet deposition and surface dry deposition. Although heterogenous $SO_2$ oxidation on ash particles, and the self-lofting effect from the ash heating effect, have both been shown to be important for the $SO_2$ dispersion from Raikoke (Muser et al., 2020) and also from the 2015 Kelud eruption (Zhu et al., 2020), ash particles are not included in these IFS simulations.*